# Cortical signatures of auditory looming bias show cue-specific adaptation between newborns and young adults
Karolina Ignatiadis [1] ✉, Diane Baier[1], Roberto Barumerli [1], István Sziller[2], Brigitta Tóth[3,4] & Robert Baumgartner [1,4] ✉

Adaptive biases in favor of approaching, or "looming", sounds have been found across ages and species, thereby implicating the potential of their evolutionary origin and universal basis. The human auditory system is well-developed at birth, yet spatial hearing abilities further develop with age. To disentangle the speculated inborn, evolutionary component of the auditory looming bias from its learned counterpart, we collected high-density electroencephalographic data across human adults and newborns. As distance-motion cues we manipulated either the sound's intensity or spectral shape, which is pinna-induced and thus prenatally inaccessible. Through cortical source localisation we demonstrated the emergence of the bias in both age groups at the level of Heschl's gyrus. Adults exhibited the bias in both attentive and inattentive states; yet differences in amplitude and latency appeared based on attention and cue type. Contrary to the adults, in newborns the bias was elicited only through manipulations of intensity and not spectral cues. We conclude that the looming bias comprises innate components while flexibly incorporating the spatial cues acquired through lifelong exposure.

One of audition's main functionalities lies in continuously monitoring our surroundings and alerting us in case of potential threats. Identifying sounds as approaching can be crucial for survival because impending objects are more likely to threaten one's own existence, primarily in an evolutionary sense[1,2]. The effect of approaching sounds being more salient than receding ones constitutes the "auditory looming bias"; a perceptual bias, presumably present to warn the sensorimotor system to take protective action. Studies corroborate this hypothesized protective nature across vertebrates: Looming sounds trigger defensive freezing and escape behaviors[3]; they moreover make animals learn faster in associative conditioning[4] and preferentially look toward the direction of the looming sound source[5,6]. Humans further exhibit this bias through faster reaction times[7], higher accuracy in motion discrimination[8], as well as overestimation of intensity changes and time to collision[9]. Due to its universal presence and ecological importance, the looming bias has been intensively studied. Although investigations have focused on younger and older humans separately, comparative studies testing those age groups on identical stimuli are needed. This lack of cross-age comparisons leaves developmental aspects of the auditory looming bias unclear.

If encoded through the evolution of species, some aspects of the looming bias may not require prior experience to be facilitated. Human newborns are presumably naive to the possibly threatening nature of looming objects and offer, at best, limited prenatal experience stemming from exposure. They, therefore, pose the best example of an unprimed human brain state that can be studied in a non-invasive manner. In fact, newborn listeners showed enhanced orientating response indicated by longer looking time, when audio-visual stimuli denote approaching motion[10]. Infants as young as four months, moreover, better discriminated against looming sounds compared to receding ones[11] and have been found to exhibit avoidance behavior when presented with them[12]. Although behavioral evidence from small samples of humans of a very young age is present, its interpretation comes with uncertainty.

Neurophysiological data can offer a complementary and more objective measure of the underlying mechanisms. Animal research revealed a crucial role of the auditory cortex in eliciting looming bias: Asymmetries in its activation reflected the looming preference[13], while its silencing inhibited looming-induced defense behaviors[14]. Human neuroimaging studies found brain areas biased in favor of looming sounds to span an extended network,

[1]Acoustics Research Institute, Austrian Academy of Sciences, Vienna, Austria. [2]Division of Obstetrics and Gynaecology, DBC, Szent Imre University Teaching Hospital, Budapest, Hungary. [3]Institute of Cognitive Neuroscience and Psychology, Research Centre for Natural Sciences, Budapest, Hungary. [4]These authors jointly supervised this work: Brigitta Tóth, Robert Baumgartner. ✉e-mail: karolina.ignatiadis@oeaw.ac.at; robert.baumgartner@oeaw.ac.at

covering temporal, parietal, and frontal cortical regions[15–17]. The specific involvement of the auditory cortex is, however, surprisingly obscure: Its appearance as an important contributing region is inconsistent across studies and raises the need for more investigations targeted towards it, under consideration of different human brain states. Apart from that, localization of the auditory cortex in neuroimaging studies is non-trivial: Although the medial part of the anatomical region of Heschl's gyrus (HG) is generally considered to host the primary auditory cortex, it remains a functional definition suffering large inter-participant variability[18].

The vast majority of previous studies on auditory looming bias moreover rely on intensity ramps as one particular cue for auditory distance motion[4,6,9–11,15,19–21]; yet sound sources moving along the distance dimension exhibit changes across multiple auditory distance cues[22]. In that context, manipulations of the sound's spectral shape have been used to elicit looming bias, thereby demonstrating that intensity ramps per se are not a necessary prerequisite[8]. Such spectral shape cues result from the acoustic filtering of an incoming sound wave by the listener's morphology, especially their pinnae. The intensity and spectral cues differ in terms of age-related exposure and the corresponding need for adaptation: HG is already developed around the 24th week of gestation[23], and fetal hearing is functional before birth. Sounds, passing through the mother's abdomen and amniotic fluid during development, undergo spectral modification and attenuation. Intensity ramps are already prenatally accessible[24], and evidence suggests that spectral information is also processed[25]. Yet newborns are additionally subject to abrupt changes in the environment postpartum. This substantially affects the characteristics of spectral cues, thereby necessitating a new acquisition or adaptation process. Both cues are known to elicit the bias under task-relevant conditions in human adults. It is, though, rather unclear, whether they also do so during inattentive listening and how they relate to each other in terms of bias characteristics and innate encoding.

In order to disentangle the speculated inborn, evolutionary component of the looming bias from counterparts potentially learned through cue-specific exposure, we collected high-density electroencephalography (EEG) data in young adults and newborns. As distance-motion cues we manipulated either the sound's intensity or spectral shape. Investigations were done at the level of the scalp as well as HG; a choice made based on prior literature[26,27] and due to it comprising the functionality of the primary auditory cortex. We hypothesized that the looming bias elicitation in young adults should be largely independent of cue type and not subject to voluntary attention in order to facilitate an effective warning mechanism. Pertinent to evolutionary processes, related aspects should be present already at the time of birth. As, though, spectral cues are highly dependent on anatomy and

familiarization, the manifestation of a spectrally induced looming bias was unexpected in newborn participants.

## Methods

### Overview

Participants were exposed to moving and static sounds presented from either the left or right side in a virtual auditory environment (Fig. 1a). Stimuli were filtered by sets of individually measured head-related transfer functions (HRTFs), namely a set of filters representing the sound modifications induced by one's pinnae, head and torso. Moving sounds differed from static sounds by having a brief cue transition phase about halfway through the stimulus (Fig. 1b, top, gray area represents the transition phase in time). The movement percept for our stimuli was created by changing either the intensity (Fig. 1b, top, blue curves) or the spectral shape (Fig. 1b, bottom, red) of a broadband harmonic tone complex. In the intensity condition, the intensity changed with time (Fig. 1b, top, blue), while the spectral content remained the same (Fig. 1b, bottom, blue), essentially representing a mere intensity offset. Spectral stimuli maintained their broadband intensity over time (Fig. 1b, top, red), but transitioned in spectral content between a flat spectrum and the measured HRTF (Fig. 1b, bottom, red). This separation was essential for our targeted dissociation between prenatally accessible intensity cues and more heavily affected, at best less accessible spectral cues. The beginning of the transition phase, hereafter referred to as the "change event" (reference point in time: 0 ms), was temporally jittered (50 ms) in order to diminish the temporal predictability of the event. The transition phase itself was kept very short (10 ms) to assure high temporal precision in the analysis of neural responses evoked by the change event. Static sounds were presented in 50% of all cases and served two purposes. First, they ensured listeners were not able to predict the stimulus category already from stimulus onset, as static sounds were constructed with the same onsets as the moving stimuli (but no transition). Second, they served as catch trials to ensure no random responses were given throughout the experiment.

We first investigated the role of attention in the elicitation of the looming bias. To this end, adult participants underwent first a passive (inattentive) and then an active (attentive) listening part. In the passive listening part, participants' attention was diverted through a silent and subtitled movie, while they were being exposed to the stimuli. During the active listening part, they performed a three-alternative motion discrimination task adapted from a previous study[8]. In it, they assessed the movement as looming, receding, or static by keyboard button press. EEG recordings from newborn participants were collected during sleep.

This study was not preregistered.

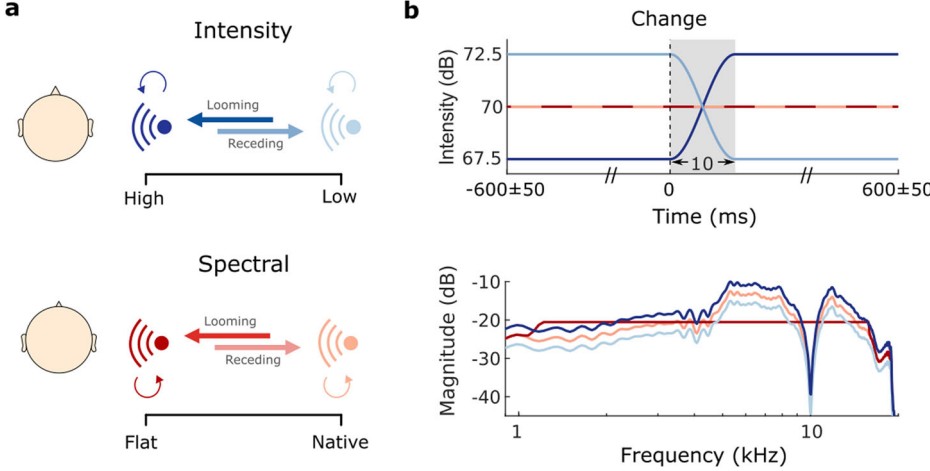

**Fig. 1 | Experimental design. a** Illustration of experimental factors movement and cue type. The transition between two sounds of different intensities (top, blue) or spectral shapes (bottom, red) creates the sensation of a moving sound source. Thick arrows represent a 50% transition probability for motion trials (dark = looming; light = receding), while thin circular arrows indicate a 50% probability for static trials. **b** Magnitude profile over time (top panel) and frequency (bottom panel) of all implemented stimuli. Filtering by the native spectral shape evokes a spatially externalized auditory percept[44]. Sounds devoid of native spectral characteristics (flat spectrum) do not elicit this externalization, making sounds appear close to one's ear.

## Adult listeners

The sample size for the adult group was determined based on the following considerations: As of Baumgartner et al.[8], 15 participants should be sufficient to detect the looming bias via scalp potentials evoked at latencies of about 160 ms for the active spectral condition. We decided to double the sample size because effect sizes were expected to be smaller under passive listening conditions, because we wanted to allow for finding neural signatures also at shorter latencies (usually of smaller amplitude and therefore harder to discern), and because we are aiming to re-use the data for exploratory connectivity studies, which generally require larger sample sizes[28].

Considering possible exclusions, we thus invited 35 healthy young adults with no self-reported indications of psychological and neurological disorders or acute or chronic heavy respiratory diseases that may prevent the participant from sitting still during the EEG recording. We initially measured participants' hearing thresholds between 1 and 12.5 kHz (AGRA Expsuite application)[29] to ensure that they deviated not more than 20 dB from their age mean[30]. Twenty-nine participants fulfilled this requirement and took part in the study. Sex and age were self-reported by the participants (15 female: 25.0 ± 2.60 years old; 14 male: 25.1 ± 2.77 years old). No data on race/ethnicity was collected. An error rate in catch-trials (static sounds) exceeding 20% was used as an exclusion criterion and resulted in one exclusion (female, 45.2% errors). Hence, $N = 28$ participants within the age range of 21–32 years were ultimately included in the study.

All participants signed informed consent prior to testing, were neither deceived nor harmed in any way, and were informed that they could abort the experiment at any time without any justification or consequences. The study was conducted in accordance with the standards of the Declaration of Helsinki (2000). No additional ethics committee approval was required given the non-medical, non-invasive nature of our study, as per the Austrian Universities Act of 2002. In total, experiments lasted around five hours per participant, and participants received monetary compensation in return for their time.

## Newborn listeners

Regarding the newborn sample size, previous event-related potential studies on neonatal auditory change detection reliably found effects with about 40 participants[31–35]. Since, due to the very specific design and paradigm, null results were to be expected in our study, we decided to substantially increase the sample size and recruit about 100 participants.

We recruited 104 healthy, full-term newborns (0–4 days after birth). Their parents provided information about the sex of the newborn participants (59 male, 45 female), as well as their birth order: 46 were firstborns, 34 were second, 14 were third child, and 6 had more than 3 siblings. None of them were twins. The mean gestational age was 40.17 ± 1 weeks, and the mean birth weight was 3787 ± 373 g. No data on race/ethnicity was collected. All newborns had normal hearing as indicated by successfully completing a Brainstem Evoked Response Audiometry (BERA) test prior to the experiment. Participant exclusion was based on the proportion of useful trials: after data pre-processing, 33 participants maintained less than 60% of the trials and were therefore excluded. Data from $N = 71$ newborns were analyzed for the present study.

Informed consent was obtained from either one or both parents. Mothers were given the choice to be present during the EEG recording; fathers were not given this choice, as, according to the hospital rules, they were only allowed to enter the ward during a daily visiting time window that did not overlap with the recording time. The study fully complies with the World Medical Association Helsinki Declaration (2000) and all applicable national laws, as approved by the National Public Health Center, Hungary.

## Stimuli

We presented harmonic tone complexes ($F_0 = 100$ Hz, bandwidth: 1–16 kHz, phase curvature: 0.5)[36] either from right or left on the horizontal plane (±90° azimuth, 0° elevation). The duration of the stimulus was 1.2 s, including on- and offset squared sine ramps of 10 ms. For moving (looming/receding) stimuli, after 600 ± 50 ms, the initial tone complex was crossfaded into the final tone complex using a linear ramp with a duration of 10 ms. Static stimuli, conversely, remained constant throughout.

The looming and receding sensations were created by two different types of spatial-distance cues, namely intensity and spectral shape (Fig. 1). The intensity manipulation resulted in a sound appearing to recede while its intensity decreased with time. We presented sounds crossfading between +2.5 dB (near position) and −2.5 dB (far position) to induce looming and receding sensations (Fig. 1a, top). For changes in spectral shape, we manipulated the individually recorded (see section Recordings in adult listeners) or semi-individualized (see section Recordings in newborn listeners) HRTFs following the procedure introduced in Baumgartner et al.[8]. The spectral shape is induced by the acoustic filtering properties of the listener's individual morphology (pinna, head, and torso) and depends on the distance and location of a sound source[37]. The highest spatial dependency is found at high frequencies. A native spectral shape reflects the characteristics of the stimulus as measured at the level of the ear canal, originating from a source positioned at a distance of 1.2 m (far position) from the listener. Flattening the spectral shape while keeping the overall intensity constant, leads to the perception of a near position (at/inside the participant's head; Fig. 1a, b, bottom).

## Procedure

Moving (looming/receding) and static trials were randomized throughout the experiment and balanced over blocks, with 50% static and 50% moving sounds. Within moving sounds, 50% were looming and 50% receding. Within static sounds, 50% corresponded to the looming stimulus onset and 50% to the receding stimulus onset. The different cue types were applied block-wise. Apart from movement and spatial cue type, we block-wise manipulated whether the sound source was presented from the left or the right side of the listener. The experimental procedures were programmed in Matlab (R2018b, Mathworks, Natick, Massachusetts) using Psychtoolbox[38].

For the adult listeners, we performed the experiments in two subsequent parts, each one under a different attentional state (passive/active). To achieve the best possible naivety, all listeners started with the passive condition. During that, they were exposed to the sounds while asked to concentrate on a muted and subtitled movie[39]. To ensure and assess that participants' attention was focused on the movie and away from the sounds, they were instructed to focus on the movie's content and informed that they would be questioned on it afterward. Performance in the subsequent testing was conceived as an exclusion criterion; yet every participant could remember the requested details from the presented documentary, leading to no participant exclusions. In active listening, the participants were tasked with discriminating the movement of the sound (looming/receding/static) by keyboard button press. For sounds presented from the right side, the left arrow key was assigned to looming, the downwards arrow key to static, and the right arrow key to receding sounds. For sounds coming from the left, the key for looming was C, for static X, and for receding Y. With this setup, the key for looming was always nearer to the participant than the key for receding sounds. Responses were permitted starting from the beginning of the crossfade ("change event", Fig. 1b). After keypress (or after the sound offset, if the response already occurred during sound presentation), an inter-stimulus interval of 800 ± 50 ms preceded the subsequent trial. During passive listening, the inter-stimulus interval was set to 500 ms. In total, the experiment comprised 1600 trials, with 100 trials per condition. Within condition, 50 trials were each presented from the left and the right side of the listener.

For the newborn listeners, the sound presentation was equivalent to the passive condition of the adults, and they underwent the experiment while in deep sleep. In contrast to active sleep, this state lasts longer (up to 60–90 min), has no rapid eye movements, and the breathing and heart rates of the newborns become more regular. Overall it is a preferable state for EEG recording, as the appearance of artifacts is much less likely[40–42]. In total, the experiment lasted approximately 30 min and consisted of 400 trials, with 100 trials per condition (as only the passive condition was considered, in

contrast to the adult experiment). Within condition, 50 trials were each presented from the left and the right side of the listeners.

## Recordings in adult listeners

We initially acquired the HRTFs for every participant individually. This was done by placing the listener in the center of a spherical array (radius of 1.2 m) of loudspeakers (E301, KEF), positioned in a semi-anechoic room (T60 = 50 ms). Two of the loudspeakers were aligned to either side of the listener's interaural axis. Small microphones (KE4-211-2, Sennheiser) were inserted in the listener's ear canals for recording. As measurement signals, we used exponential sweeps ranging from 20 Hz to 20 kHz within 6 s. Sweeps were multiplexed across directions in order to speed up the whole measurement duration and thus minimize the risk of artifacts introduced by small movements of participants[43]. The acoustic influence of the equipment was removed by equalizing the HRTFs with the transfer functions of the equipment. Those were derived from prior reference measurements, during which the in-ear microphones were placed at the center point of the spherical loudspeaker array in the absence of the listener. The measured listener-specific HRTFs were then used to filter the presented stimuli. This individualized filtering procedure creates the impression of virtual sound sources in space when presented via headphones[44]. To verify our HRTF measurement, prior to the actual experiment, we introduced the listeners to three horizontal sound trajectories[45], that started in front of them and moved in a circle around their heads twice. Each of the trajectories was filtered with either their own or one of two arbitrarily chosen non-individual HRTFs. Participants could listen to the different trajectories as often as they wanted before choosing the trajectory that felt most natural to them. Over half of the participants (53.6%) consistently chose the trajectory filtered with their own HRTF set. 17.9% consistently chose a different HRTF, and the remaining participants made inconsistent choices; both could occur due to coincidental similarities between their own and a non-individual HRTF set. For the main experiment, the individually measured HRTF set was replaced by the non-individual HRTF set only if the participant consistently preferred that set during the verification process.

To record scalp activity, we used a 128-channel EEG system (actiCAP with actiCHamp; Brain Products GmbH, Gilching, Germany) and recorded at a sampling rate of 1 kHz. For sound presentation, participants wore ER-2 insert earphones (Etymotic Research Inc., Grove Village, Illinois). After concluding the experiment, we made an optical 3D scan of the electrode positions using the Structure Sensor with Skanect Pro (Occipital Inc., Boulder, Colorado). Adult experiments took place at the Acoustics Research Institute of the Austrian Academy of Sciences.

On a different day, a structural T1-weighted scan was recorded at the MR center of the SCAN-Unit (Faculty of Psychology, University of Vienna) with a 3 Tesla magnetic resonance imaging system (MRI; 32-channel head coil; Siemens MAGNETOM Skyra, Siemens-Healthineers, Erlangen, Germany). Structural images were acquired using a magnetization-prepared rapid gradient-echo sequence with the following parameters: TE = 2.43 ms; TR = 2300 ms; 208 sagittal slices; field-of-view: 256 × 256 × 166 mm; voxel size: 0.8 × 0.8 × 0.8 mm.

## Recordings in newborn listeners

Since HRTF measurements are not feasible with newborn listeners, we used a combination of two anthropometric measures to individualize a template HRTF by means of frequency scaling[46]. One metric denotes the pinna-cavity height as measured from the inter-tragal notch to the rim of the helix. The other metric denotes the head width as measured from side to side at the point in front of the tragus that is defined by the condyle of the mandible. As the template HRTF, we selected one from the institute's public database (NH92)[47] with a pinna-cavity height of 44 mm and a head width of 134 mm. To record brain activity, we used a 65-channel EEG system (R-Net with actiCHamp; Brain Products GmbH, Gilching, Germany) and recorded at a sampling rate of 500 kHz. A 100 Hz online low-pass filter was applied. Electrodes were placed according to the International 10/20 system. The Cz channel served as the reference electrode, while the ground electrode was

placed on the midline of the forehead. During the recording, impedances were kept below 15 kΩ. Stimuli were presented using an external sound card (Maya22 USB, ESI Audiotechnik GmbH, Leonberg, Germany) with ER-2 Insert Earphones (Etymotic Research Inc., Elk Grove Village, IL, USA) placed into the newborns' ears via ER2 Foam Infant Ear-tips. EEG was recorded throughout the stimulus presentation. Newborn experiments took place at the Department of Obstetrics-Gynecology, Szent István Hospital, Budapest, Hungary.

Newborn participants were asleep for the duration of the stimulus presentation. Sleep state was determined based on standardized behavioral criteria[48]. Only participants that were in quiet sleep for the whole 35-min duration of the experiment were included in the study. In addition to the behavioral criteria employed, the EEG signal was visually inspected, to ensure muscle tension was tonic, respiration regular and eye movements absent.

## Behavioral analysis

To investigate the presence of the looming bias behaviorally in the adult listener pool, we jointly analyzed choice and response time data by using a linear ballistic accumulator model[49], as it provides a tractable analytical solution for multiple conditions[50]. The model's design considers a multi-alternative response time task, where each possible response competes against the others by accumulating information with a specific speed $v_i$, termed the drift rate. Each accumulator starts from a random point, sampled from the uniform distribution $[0, A]$. The first accumulator to reach the threshold $b$ determines a participant's response. A non-decision time $t_0$ is added to the time of threshold exceedance, accounting for the remaining non-specific variance (e.g., motor latency). We used the hierarchical Bayesian implementation of the linear ballistic accumulator, to study parameter changes at a group level[50,51]. Via this approach, the estimation procedure could rely on fewer trials while accounting for between-participant variability. For the parameter estimation, we chose the differential evolution Markov Chain Monte Carlo (DE-MCMC) sampling[50,52], which accounts for the correlation among free parameters.

We considered the moving trials (looming and receding) and clustered them by cue type (spectral and intensity) and response correctness (correct and incorrect). The model framework instantiates one accumulator per condition and response choice. Based on our design, we fitted 8 accumulators per participant. This configuration led to 11 parameters per participant, of which 8 represented drift rates per condition, and the 3 remaining parameters, namely the starting interval, threshold, and non-decision time, were shared across conditions. Starting points for the Markov chains were drawn according to the following normal distributions truncated to only allow for positive values: $A \sim N(2, 0.2)$, $b \sim N(1, 0.1)$, drift rates for correct responses $v_c \sim N(3, 0.3)$ and for incorrect responses $v_e \sim N(1, 0.1)$, and $t_0 \sim N(0.2, 0.02)$. Due to the hierarchical settings, the participant-level parameters depended on the group-level truncated normal distribution with its own mean and standard deviation. Priors of these group-level parameters were sampled from truncated normal distributions, with $A_\mu \sim N(2, 1)$, $b_\mu \sim N(2, 1)$, drift rates for correct responses $v_{c\mu} \sim N(3, 1)$ and incorrect responses $v_{e\mu} \sim N(1, 1)$, and $t_{0\mu} \sim N(0.2, 0.1)$. Standard deviation parameters were defined as gamma distributions with both shape and scale parameters set to 1, except for $t_0$, for which the scale parameter was set to 3. The choice of those priors was based on previous design propositions[50,51]. To account for the difference in experimental procedures, we here doubled the overall number of samples and tripled the burn-in length. As a result, the fitting procedure used 32 interacting Markov chains, each with a length of 8000 samples. Six thousand out of those were burn-in samples, and a thinning of 5 samples was applied to the remaining ones. Thinning was introduced to reduce the amount of autocorrelation. To assess the convergence of the MCMC, we relied on the Gelman-Rubin diagnostic, which returned a mean value of 1.006 ± 0.003 (max 1.015)[53]. Our parameter fitting procedure returned the following means and standard deviations for the shared parameters at a group level: $A = 0.573 \pm 0.720$ s, $b = 1.694 \pm 0.521$ s, $t_0 = 0.139 \pm 0.208$ s.

We additionally evaluated the ability of the model to replicate the actual data by running posterior prediction checks for each condition, assessed by computation of the two-sided p-value and 95% credible intervals[54]. For each participant, we randomly drew 50 samples from each chain. As test statistic, we considered the proportion of simulated response times falling within the first and third quartiles of the corresponding values for the actual data. The same procedure was followed for the simulated response accuracies.

To finally assess the difference in drift rates between the looming and receding conditions, we sampled the mean and variance of the drift rates from the posterior distributions at the group level. We used these parameters to characterize a Gaussian distribution, from which we generated $N = 10,000$ samples per motion direction. In order to quantify the looming bias, we defined the ratio of samples indicating a higher drift rate for looming than receding, relative to the total number of samples: $r = N^{-1} \sum_{i=0}^{N} \mathbf{1}_{\mathbb{R}^+}(v_{L,i} - v_{R,i})$, where $v_{L,i}$ denotes a sampled drift rate for looming, $v_{R,i}$ for receding and $\mathbf{1}_{\mathbb{R}^+}(\cdot)$ represents the indicator function returning one for strictly positive values, zero otherwise. We repeated this procedure 10,000 times to compute the probability of observing a ratio larger than the chance level (i.e., 0.5) using a one-tailed 89% credible interval[55]. We finally computed the ratio separately for each cue type.

For the above analysis, we used R (R Core Team, 2023) with the packages: `data.table`[56], `msm`[57], `coda`[58] and `ggplot2`[59].

## Adult EEG analysis

EEG data were visually inspected to single out potential bad channels, which were then interpolated. The data were subsequently bandpass-filtered between 0.5 and 100 Hz (Kaiser window, $\beta = 7.2$, $n = 462$) and epoched to stimulus onset ([−200, 1500] ms). A hard threshold of −200 to 800 μV was additionally applied, to detect trials that still had large outlier values, potentially denoting issues that went undetected by visual inspection (e.g., excessive movement artifacts, intermittently broken channels)[8,26]. A further step for automatic channel rejection was used to detect potentially undetected noisy channels. If found, they would next be visually inspected and interpolated. No additional noisy channels were detected for any of the participants. We performed independent component analysis (ICA) decomposition and followed up with a manual artifact inspection and rejection of oculomotor artifacts (up to 3 components removed per participant). The data were thereafter re-referenced to their average and re-epoched to the change event ([−550, 850] ms). The channel positions were subsequently overwritten by the individual ones, which had been acquired by manually tagging them on the 3D head scans we recorded after each experiment. Trials were equalized within each participant to match the minimum amount within the participant after trial rejection, aiming at an equal distribution across the recordings. More specifically, we selected every $(y/x)^{th}$ trial in order to remove $x$ trials from a set of $y$ trials, a process rendering the same amount of trials across conditions within a participant. On average, this resulted in $92 \pm 4.6$ trials per participant and condition. Scalp ERPs were additionally low-pass filtered at 20 Hz (Hamming-based FIR, $n = 150$) with ERPLAB[60] and baseline-corrected by a 100-ms-pre-event interval. We deliberately did not apply this low-pass filtering directly at the beginning; that way, our initial filtering (0.5–100 Hz) still allows for later exploratory analyses on an extended frequency range. All steps were undertaken in EEGLAB[61] as well as custom Matlab scripts.

Anatomical MRIs for all participants were segmented via Freesurfer, version 7.1.1[62], and used to create a study protocol on Brainstorm[63]. For three of the participants, the default anatomical models of Brainstorm were used (ICBM152 brain template), as we could not acquire individual MRIs due to incompatibilities with the scanner (suspicion of metallic parts in the body). Anatomical models were created via OpenMEEG[64]: for the boundary element model (BEM) surfaces we used 1922 vertices per layer for the scalp, outer skull, and inner skull, and a skull thickness of 4 mm. The relative conductivity was set to 0.0125 for the outer skull and to 1 for the remaining layers. For each participant, we performed a manual co-registration between the head models and the individual channel locations. To infer cortical source activity, we used the dynamic statistical parametric mapping (dSPM)

inverse solution[65], based on previous investigations showing better HG localization performance compared to standardized low-resolution electromagnetic tomography (sLORETA)[66]. For that, the noise covariance was calculated from a 200 ms pre-stimulus interval, the source orientations were considered constrained, and source signals were reconstructed at 15000 vertices describing the pial surface. For consistency and comparability with previous relevant literature[26,27], evoked HG activity was extracted according to the Desikan–Killiany parcellation scheme as defined in Brainstorm (transverse temporal region)[67].

Amplitudes and latencies of the N1 and P2 components were extracted based on the individual averaged time courses of the participants and the function `findpeaks` (Matlab R2018b, Mathworks, Natick, Massachusetts). Since we already low-pass filtered the data at 20 Hz, we deliberately opted against additional low-pass filtering through some form of temporal averaging and simply took the amplitude and latency of peaks identified within certain time windows that are consistent with values of N1 and P2 latencies reported in the literature. We set the time windows in which to search for the components, after careful inspection of all individual ERP profiles in order to ensure no local minima or fluctuations affected our results. Considering literature values and adapting the intervals after visual inspection, for the scalp ERPs, the N1 component peak was considered within the time window from 82 to 182 ms after the change event. For the source analysis, this window was placed slightly earlier, from 77 to 177 ms. The P2 component peak was defined in a case-specific manner: starting at the timing of each individual N1 peak, the P2 peak was searched within a subsequent window of 150 ms. In cases where no peaks could be found, such as for poor source localization or untypical scalp timeseries profiles lacking peaks, the corresponding participants were not considered in the statistical analyses (concerned 2 participants each for scalp P2, source N1, and source P2). We opted for that solution as it was deemed a more objective one, compared to arbitrarily assigning a peak value based on literature values or participant means.

[68]. Repeated-measures ANOVAs were done after testing for sphericity (Mauchly's W) and normality (Q-Q plot) of our data in R (R Core Team, 2023). For the assessment of statistical differences in the time series, we used a cluster-based permutation test implemented in FieldTrip (`ft_timelockstatistics`)[69]: We assessed the p-value via 500 Monte Carlo permutations and implemented a two-tailed t statistic ($\alpha = 0.05$) on the samples, which then summed up within a cluster to form the cluster-level values. As our cluster-level metric, we used the maximum of the cluster-level statistics in a permutation test ($\alpha = 0.05$). The effect size was assessed by means of Cohen's d (`meanEffectSize` implemented in Matlab). An additional Bayesian repeated-measures ANOVA performed on the onset scalp-ERPs considered the factors of attention (active or passive), cue type (intensity or spectral), and position (near or far) in a $2 \times 2 \times 2$ design. To that end, we averaged the corresponding onset scalp time series (vertex electrode Cz) across the time interval between 0 and 200 ms, in order to capture potential effects linked to the sound onset. We investigated the effects across matched models using default settings ($r$ scale fixed effects = 0.5, $r$ scale random effects = 1, $r$ scale covariates = 0.35). This analysis was implemented in JASP, version 0.17.3 [68].

## Newborn EEG analysis

Data were highpass-filtered at 0.05 Hz (Hamming window, $n = 33,000$)[70] and lowpass-filtered at 80 Hz (Hamming window, $n = 84$). Compared to the adults, we chose the highpass cutoff frequency much lower for the newborns to ensure the inclusion of the slow oscillations that are typical for neonate brains[70,71]. After visual inspection, noisy channels were singled out and interpolated (maximally 5 per participant) using the default spline interpolation algorithm implemented in EEGLAB [61]. We next re-referenced our data to their average and epoched them ([−100, 800]) time-locked to the change of the stimulus (beginning of the cross-fade, Fig. 1b, top). A baseline correction using a 100-ms-pre-event interval was performed. A hard threshold of −100 to 100 μV was additionally applied, to detect large outlier trials. Data were finally visually inspected, and noisy epochs were manually

removed. Trial numbers were equalized across conditions within each participant by removing trials equally distributed across the recordings, in order to match the minimum amount within the participant. Participants with less than 60% of the trials per condition were excluded from the study. This process resulted, on average, in 82 ± 8.5 trials per participant in every condition. Scalp ERPs were low-pass filtered at 20 Hz (Hamming-based FIR, $n = 140$) with ERPLAB [60]. All pre-processing steps were undertaken in the EEGLAB [61] free software as well as custom Matlab scripts.

Identically to the adults (Adult EEG analysis), statistical differences in the scalp topographies were assessed by cluster-based permutation testing. The effect size was in all cases assessed by means of Cohen's $d$ (`mean-EffectSize`, Matlab). In contrast to the EEG recordings in the adult group, Cz was used as a reference during newborn recordings. Following the common practice of infant ERP analysis, a cluster of channels was considered to estimate the effects [72]. We calculated the scalp-ERPs based on the emerging frontocentral cluster of electrodes, comprising electrodes *Fp1, AF7, AF3, AFz, AF4, F5, F3, F1, Fz, F2, F4, F6, FC3, FC1, FC2, FC4, C1, C2, and C4*.

In the case of the onset ERPs, an additional Bayesian repeated-measures ANOVA was done, with the factors of cue type (spectral or intensity) and position (near or far) in a $2 \times 2$ design. To that end, the corresponding time series were averaged across the time interval of 0–200 ms, considered to capture the onset-locked responses of the stimuli. As the time interval of choice was, to a degree, arbitrary, we repeated the analysis by considering the data over the longer time interval of 0–400 ms. Changing the time interval did not change our null results in the onset analyses. This analysis was implemented in JASP, version 0.17.3 [68].

For the anatomical modeling we replicated the process followed in the adult data analysis, with the following differences: In the absence of individual MRIs, template anatomical models implemented in Brainstorm ('Oreilly' 0.5 month brain template) were used, fitted with the default channel cap adjusted to our electrode configuration. The relative conductivity of the outer skull was set to 0.0041 and to 0.33 for the remaining layers [73].

An additional Bayesian repeated-measures ANOVA considered the factors of motion (looming or receding) and hemisphere (left or right) in a $2 \times 2$ design for the spectral condition. To that end, we averaged the corresponding HG time series across the time interval between 250 and 450 ms in the looming as well as receding spectral cue time series. As there were no specific peaks that would allow us to exactly follow the statistical process we followed in the adult data, we chose this time window as representative of the looming bias activation, based on the significant clusters found for the intensity condition. We investigated the effects across matched models using default settings ($r$ scale fixed effects = 0.5, $r$ scale random effects = 1, $r$

scale covariates = 0.35). As the choice of this time interval is to some degree arbitrary, we performed robustness tests by repeating the same procedure for an earlier time interval (200–400 ms), as well as for the latest interval of 600–800 ms, qualitatively showing the biggest deviation between the looming and receding time series. Changing the considered time windows did not change our results. This analysis was implemented in JASP, version 0.17.3 [68].

## Reporting summary
Further information on research design is available in the Nature Portfolio Reporting Summary linked to this article.

## Results
### Behavioral results: looming sounds speed up evidence accumulation
The adult participants detected static sounds very accurately (hit rates: [0.955, 0.984, 0.995], denoting 25%, 50%, and 75% percentiles) and quickly (response times for hits: 0.944 ± 0.114 s, denoting mean ± standard deviation) throughout the entire active task. This high performance on catch trials confirmed our listeners were attentive. When comparing static sound detection with the Wilcoxon Signed-Rank test, the discrimination of movement direction in motion trials was substantially harder (hit rates: [0.514, 0.648, 0.757], $V = 406$, $N = 27$, $p < 0.001$) and slower (response times: 1.017 ± 0.170 s, $V = 36$, $N = 27$, $p < 0.001$). Given the almost perfect hit rates for catch trials, we simplified subsequent analyses by only considering the motion trials (as a two-alternative forced-choice task).

Figure 2a reports the behavioral measures of accuracy and response time across conditions. To identify differences between motion direction and cue type, we fitted a hierarchical linear ballistic accumulator model with a differential evolution Markov chain Monte Carlo (MCMC) method [52]. We selected this model-based approach because of its advantage in accounting for the speed-accuracy trade-off on a trial-by-trial level [74] as well as the different uncertainty levels across participants [50]. In this modeling framework, an evidence accumulation process is started for every choice option and trial; the accumulator hitting the response threshold first decides the choice as well as the response time. To study the presence of looming bias, our latent variable of interest was the drift rate, which quantifies the velocity of evidence accumulation towards a response in a forced-choice task [49]. With drift rates fitted for every stimulus condition, the comparison between simulated and measured data revealed high agreement since the difference between actual and simulated hit rates (diff = −0.005, 95% CI [−0.110, 0.110]) and the difference in inter-quartile range of response times (diff = −0.020 s, 95% CI [−0.194, 0.358] s) showed no statistically significant evidence for deviation from zero (i.e., there is no statistical

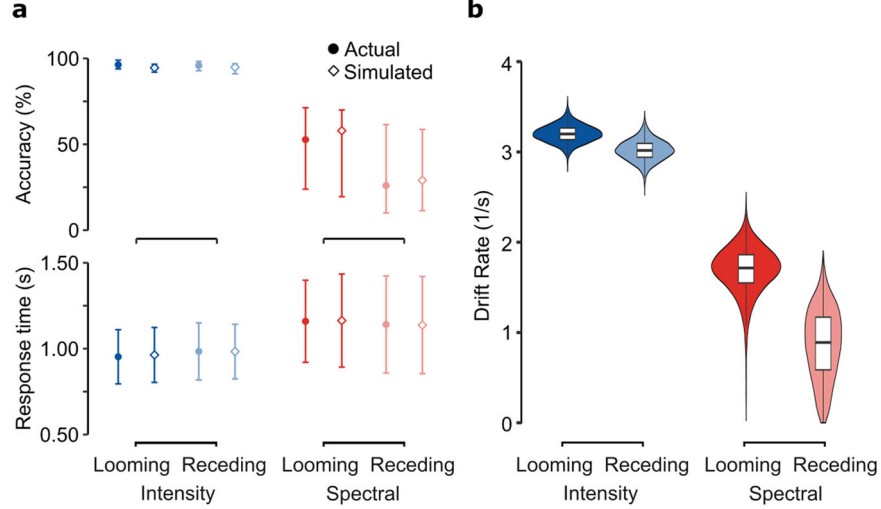

**Fig. 2 | Model-based analysis of adults' behavioral responses indicates speed-up of evidence accumulation. a** Response times and accuracies contrasted between actual data and simulated responses generated by a linear ballistic accumulator model with fitted group-level parameters. Symbols denote means for response times and medians for accuracies. Error bars denote the standard deviation for response times and the first and third quartiles for accuracies. **b** Posterior distributions of drift rate estimates indicating the listeners' speed of evidence accumulation for correctly discriminated motion directions. Center lines show medians, box limits show interquartile ranges, and whiskers show ranges up to 1.5 times the interquartile range. $N = 28$.

difference since the confidence interval includes zero, see Fig. 2a). Figure 2b shows the corresponding posterior distributions of the drift rate estimates at the group level. Most importantly, drift rates turned out higher for looming than receding sounds, as confirmed by the ratio of larger drift rates sampled from the posterior distributions when aggregating over different spatial cues ($r = 0.640$, 95% CI [0.518, 0.749], $p(r > 0.5) = 0.986$) and when considering the intensity ($r = 0.598$, 95% CI [0.447, 0.739], $p(r > 0.5) = 0.896$) and spectral condition ($r = 0.684$, 95% CI [0.485, 0.837], $p(r > 0.5) = 0.966$) separately.

### Adults' change-evoked scalp potentials: looming bias elicited during passive listening

We next investigated the looming bias by analyzing the EEG responses at the scalp. Following prior literature[8], we extracted our signals from the vertex electrode (Cz), a choice we subsequently validated through topographic analyses across the scalp. On average, across looming and receding trials, the change events evoked larger scalp potentials during the active auditory task engagement as compared to passive auditory exposure (Fig. 3a). Auditory-evoked responses displayed stereotypical N1 and P2 components and were higher in amplitude for spectral than intensity cues.

For the evaluation of the looming bias, we computed the difference between looming and receding trials (*looming − receding*; Fig. 3b). To investigate the scalp distribution and timing of emerging biases, we performed a cluster-based permutation test[69] on the temporal evolution of scalp topographies. The emerging profile is consistent among all conditions and manifested as a significant central spatial cluster (Fig. 3e): For each cue type, in the passive condition, statistically significant looming bias cluster peaks were found around 120 ms (passive spectral: 112 ms,

clusterstat $= -4.701 \times 10^3$, $p = 0.010$, $d = 0.660$, 95% CI [0.352, 1.012]; passive intensity: 146 ms, clusterstat $= -9.475 \times 10^3$, $p = 0.004$, $d = 1.175$, 95% CI [0.726, 1.761]), while no statistically significant evidence of a difference emerged at the later stages of auditory processing. In the active cases, significant clusters emerged later for both cue types (active spectral: 197 ms, clusterstat $= 1.023 \times 10^4$, $p = 0.002$, $d = 0.807$, 95%-CI [0.521, 1.298]; active intensity: 241 ms, $2.982 \times 10^4$, $p = 0.002$, $d = 1.313$, 95% CI [0.842, 1.847]). While no statistically significant evidence of a bias cluster was found in the earlier time window for the active spectral condition, a bias cluster emerged as significant for the active intensity condition, at 150 ms (clusterstat $= -2.559 \times 10^4$, $p = 0.002$, $d = 1.949$, 95% CI [1.418, 2.701]). The time point of maximum bias manifestation within the clusters differed with cue type and attentive state; within the active state, the maximum bias appeared 44 ms later for intensity cues than spectral cues and 34 ms later in the corresponding passive conditions.

For further statistical comparison of the factors cue type and attention, we extracted the peak amplitudes (Fig. 3c) and corresponding latencies (Fig. 3d) of the N1 and P2 components for all considered conditions at the vertex electrode (Cz) site; placed centrally in the emerging topographies, it is considered representative of the significant topographic clusters. Our analyses revealed significant effects of cue type on N1 and P2 amplitudes and latencies. The components' peaks appeared larger and later for intensity cues compared to spectral ones. Attention showed little effect on N1 peaks but significantly magnified P2 biases, especially for intensity cues.

Specifically, for the N1 component, significant differences in amplitude ($F(1, 27) = 4.199$, $p = 0.05$, $\eta_G^2 = 0.053$, 95% CI [0.00, 1.00]) and latency ($F(1, 27) = 18.99$, $p < 0.001$, $\eta_G^2 = 0.188$, 95% CI [0.02, 1.00]) were found only between the cue types. The amplitude bias was larger (diff $= 0.438 \mu V$,

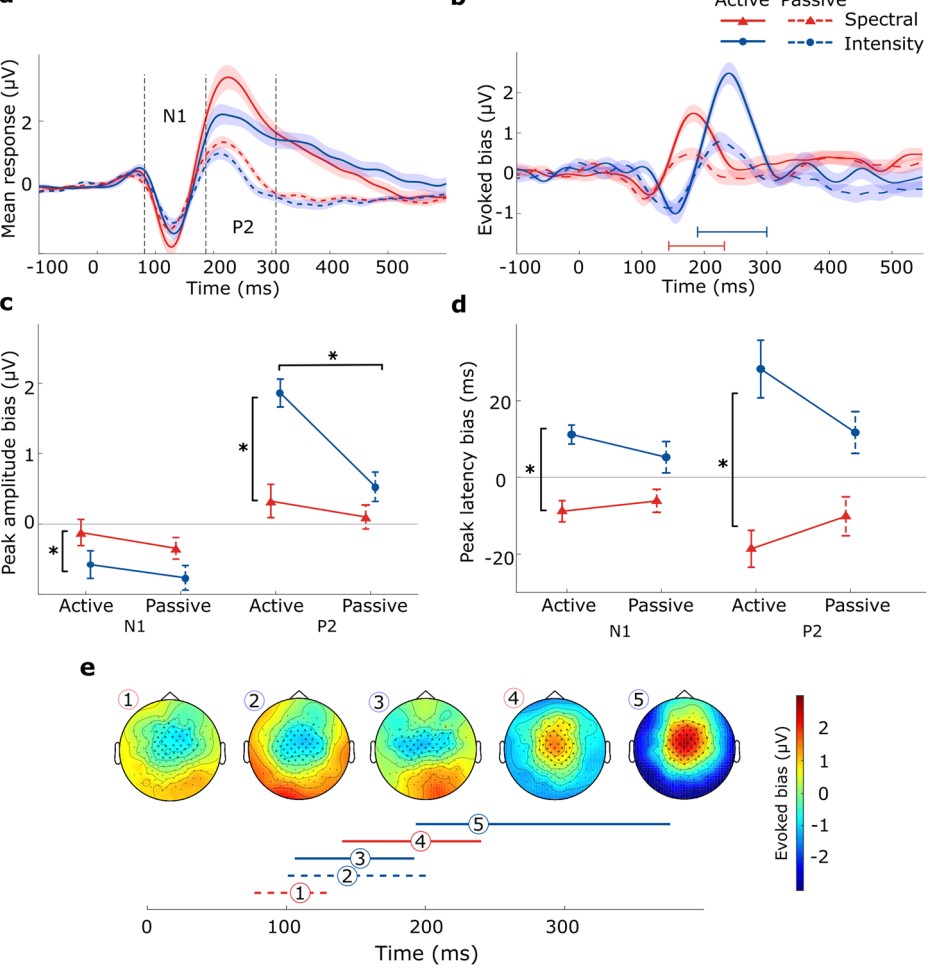

**Fig. 3 | Adults' change-evoked scalp potentials reveal auditory looming bias across attentional states and cue types. a** Potentials evoked at the vertex electrode (Cz) on average across looming and receding trials (*looming*/2 + *receding*/2). Shaded areas denote the standard errors of the means. **b** Difference waveforms (*looming − receding*) at the vertex electrode. **c** Extracted peak amplitude values of the N1 and P2 components. Error bars represent 95% confidence intervals. Asterisks indicate significant main effects ($p < 0.05$) per component. **d** Extracted peak latency values of the N1 and P2 components. **e** Scalp topographies and duration of clusters with significant looming bias, defined as the difference between looming and receding trials. Horizontal lines denote the durations of the significant clusters and are tagged with numbers at the point of maximum manifestation. $N = 28$.

**Fig. 4 | Adult participants' ERPs locked to sound onset show no differences between near and far distances. a** Grand-average topographic maps around N1 and P2 deflections (top) and evoked Cz potentials (bottom) depending on attention, averaged over cue type and distance. **b** Comparisons of evoked Cz potentials between distances within cue type. Shaded areas denote standard errors of means $N = 28$.

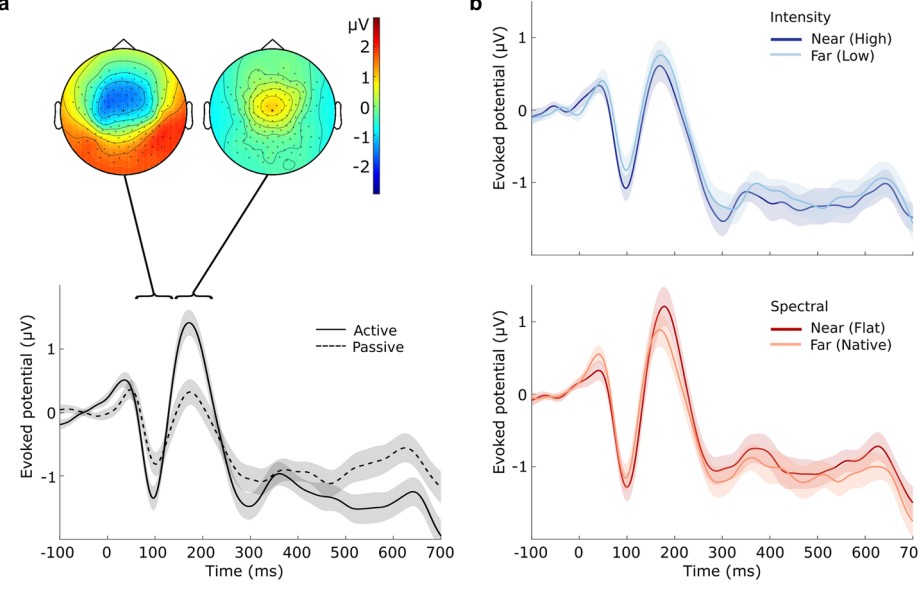

$t(27) = 2.049$, $p = 0.05$, $d = 0.463$, 95% CI [0.01, 0.93]) and occurred later (diff = 0.016 s, $t(27) = 4.359$, $p < 0.001$, $d = 0.944$, 95% CI [0.51, 1.38]), for intensity than for spectral cues. For the P2 component, we found a significant main effect of the attentional state on peak amplitude biases ($F(1, 25) = 22.51$, $p < 0.001$, $\eta_G^2 = 0.114$, 95% CI [0.00, 1.00]), with larger biases for active than passive listening (diff = 0.752 $\mu$V, $t(25) = 4.744$, $p < 0.001$, $d = 0.703$, 95% CI [0.34, 1.06]). For cue type, peak amplitudes ($F(1, 25) = 12.77$, $p = 0.001$, $\eta_G^2 = 0.174$, 95% CI [0.01, 1.00]) and peak latencies ($F(1, 25) = 19.20$, $p < 0.001$, $\eta_G^2 = 0.231$, 95% CI [0.04, 1.00]) turned significant, with larger (diff = 0.961 $\mu$V, $t(25) = 3.574$, $p = 0.001$, $d = 0.899$, 95% CI [0.33, 1.47]) and later (diff = 0.034 s, $t(25) = 4.382$, $p < 0.001$, $d = 1.076$, 95% CI [0.58, 1.57]) biases for intensity than spectral cues. We moreover found a significant interaction between the attention and cue type factors (amplitude: $F(1, 25) = 5.54$, $p = 0.027$, $\eta_G^2 = 0.055$, 95%-CI [0.00, 1.00]; latency: $F(1, 25) = 5.072$, $p = 0.033$, $\eta_G^2 = 0.051$, 95%-CI [0.00, 1.00]): amplitude values for active intensity looming bias were higher than those for passive (diff = 1.258 $\mu$V, $t(25) = 4.071$, $p < 0.001$, $d = 1.177$, 95% CI [0.37, 1.99]), and only within the active condition, intensity looming biases were larger (diff = 1.468 $\mu$V, $t(25) = 4.262$, $p < 0.001$, $d = 1.373$, 95% CI [0.36, 2.39]) and more delayed (diff = 0.048 s, $t(25) = 4.815$, $p < 0.001$, $d = 1.531$, 95% CI [0.67, 2.39]) than those for the spectral condition.

In order to check for potential distance-specific effects evoked by the starting positions of the sounds, we replicated the above temporal cluster-based permutation analysis for the neural signatures locked to the sounds' onsets (Fig. 1b top, timepoint $-600 \pm 50$ ms). Within cue type, we compared responses to sounds representing a near versus far distance from the listener (spectral: flat vs. native; intensity: high vs. low). Adult listeners exhibited ERPs with central topographies and stereotypical deflections (at vertex electrode Cz) magnified through attention (Fig. 4a). Paired comparisons evaluated by means of cluster-based permutation testing revealed no statistically significant evidence for differences between near and far distances within cue type (Fig. 4b), hinting at a null effect of the simulated starting position of each sound. A $2 \times 2 \times 2$ Bayesian repeated-measures ANOVA with the factors attention (active or passive), cue type (spectral or intensity) and position (near or far) was performed. Bayes factor for exclusion (analysis of effects) for all factors, as well as their interaction, yielded no reliable evidence for or against a positional bias (attention: $BF_{excl} = 0.011$; cue type: $BF_{excl} = 4.830$; position: $BF_{excl} = 1.431$; attention × cue type: $BF_{excl} = 0.926$; attention × position: $BF_{excl} = 1.983$; cue type × position: $BF_{excl} = 2.427$; attention × cue type × position: $BF_{excl} = 3.437$).

## Adults' source activity: early preattentive bias in Heschl's gyrus

Based on individual brain anatomies and recorded electrode locations, we inferred the recorded activity on the cortical surface[66]. The change events evoked neural activity strongly focused on the targeted HG (Fig. 5a). Both the left and right HG exhibited stereotypical auditory evoked responses for all considered conditions. In addition, we found high activity at more posterior regions (planum temporale), while activations seem to have leaked into the posterior regions of the insular cortex. Further investigation of these ROIs outside HG was out of the scope of the current study.

As done at the scalp level, we investigated the looming bias as the difference between looming- and receding-evoked source activity (Fig. 5b). In both cortices, we observed qualitatively similar waveforms, that were also congruent to the scalp responses (Fig. 3b). Cluster-based permutation tests revealed a significant looming bias for all conditions bilaterally (for the clusters in order of appearance over time; HG left: active intensity: clusterstat = 249.13, $p < 0.001$, $d = 3.91$, 95% CI [3.28, 4.69], passive intensity: clusterstat = 98.45, $p = 0.004$, $d = 4.18$, 95% CI [3.29, 5.33] and clusterstat = 79.06, $p = 0.02$, $d = 4.26$, 95% CI [3.29, 5.58] active spectral: clusterstat = 97.67, $p < 0.001$, $d = 0.55$, 95% CI [0.43, 0.69], clusterstat = 47.31, $p = 0.003$, $d = 1.57$, 95% CI [1.10, 2.28] and clusterstat = 98.09, $p < 0.001$, $d = 1.21$, 95% CI [0.97, 1.55]; HG right: active intensity: clusterstat = 119.54, $p < 0.001$, $d = 0.99$, 95% CI [0.80, 1.23] and clusterstat = 171.02, $p < 0.001$, $d = 5.14$, 95% CI [4.22, 6.29]; passive intensity: clusterstat = 168.44, $p < 0.001$, $d = 1.54$, 95% CI [1.28, 1.87] and clusterstat = 103.38, $p = 0.005$, $d = 2.11$, 95% CI [1.67, 2.69]; active spectral: clusterstat = 147.96, $p < 0.001$, $d = 0.68$, 95% CI [0.55, 0.84] and clusterstat = 120.82, $p < 0.001$, $d = 0.78$, 95% CI [0.63, 0.98]), with the exception of the passive spectral condition, which only elicited the bias in the right HG (clusterstat = 57.95, $p = 0.02$, $d = 2.33$, 95% CI [1.72, 3.24]; Fig. 5b, right).

Deflections representing the N1 and P2 components were used to more systematically investigate the considered factors of attention and cue type. We extracted peak amplitude values and latencies for those components and quantified the bias as the difference between the looming- and the receding-evoked activity (Fig. 5c).

Looming bias in the N1 amplitude depended on cue type ($F(1, 25) = 6.15$, $p = 0.02$, $\eta_G^2 = 0.040$, 95% CI [0.00, 1.00]), reflecting larger biases for the intensity compared to the spectral condition (diff = 0.039 $\mu$V, $t(25) = 2.479$, $p = 0.02$, $d = 0.4$, 95% CI [0.06, 0.75]). For P2 amplitudes, main effects were found not only for cue type ($F(1, 25) = 4.77$, $p = 0.038$, $\eta_G^2 = 0.027$, 95% CI [0.00, 1.00]) but also for attention ($F(1, 25) = 10.12$, $p = 0.004$, $\eta_G^2 = 0.086$, 95% CI [0.00, 1.00]): biases were stronger for

**Fig. 5 | Change-evoked activity in HG of the adult participants reveals auditory looming bias across attentional states and cue types. a** Evoked activity, averaged over looming and receding trials, for left HG (left figure column) and right HG (right column), including lateral views of whole-brain source activations at 120 ms (N1 peak). Blue contours within the brain maps indicate the borders of the target ROIs (HG). Shaded areas depict the standard errors of the means. **b** Looming bias (looming − receding) evoked activity for left HG (left) and right HG (right). Horizontal lines denote the durations of significant temporal clusters. **c** Peak N1 and P2 amplitude values for evoked HG activity depending on brain hemisphere, type of cue (intensity/spectral shape changes), and attentional state (active/passive). Error bars represent 95% confidence intervals. Asterisks indicate significant main effects ($p < 0.05$) per component. **d** Peak N1 and P2 latency values for evoked HG activity. $N = 28$.

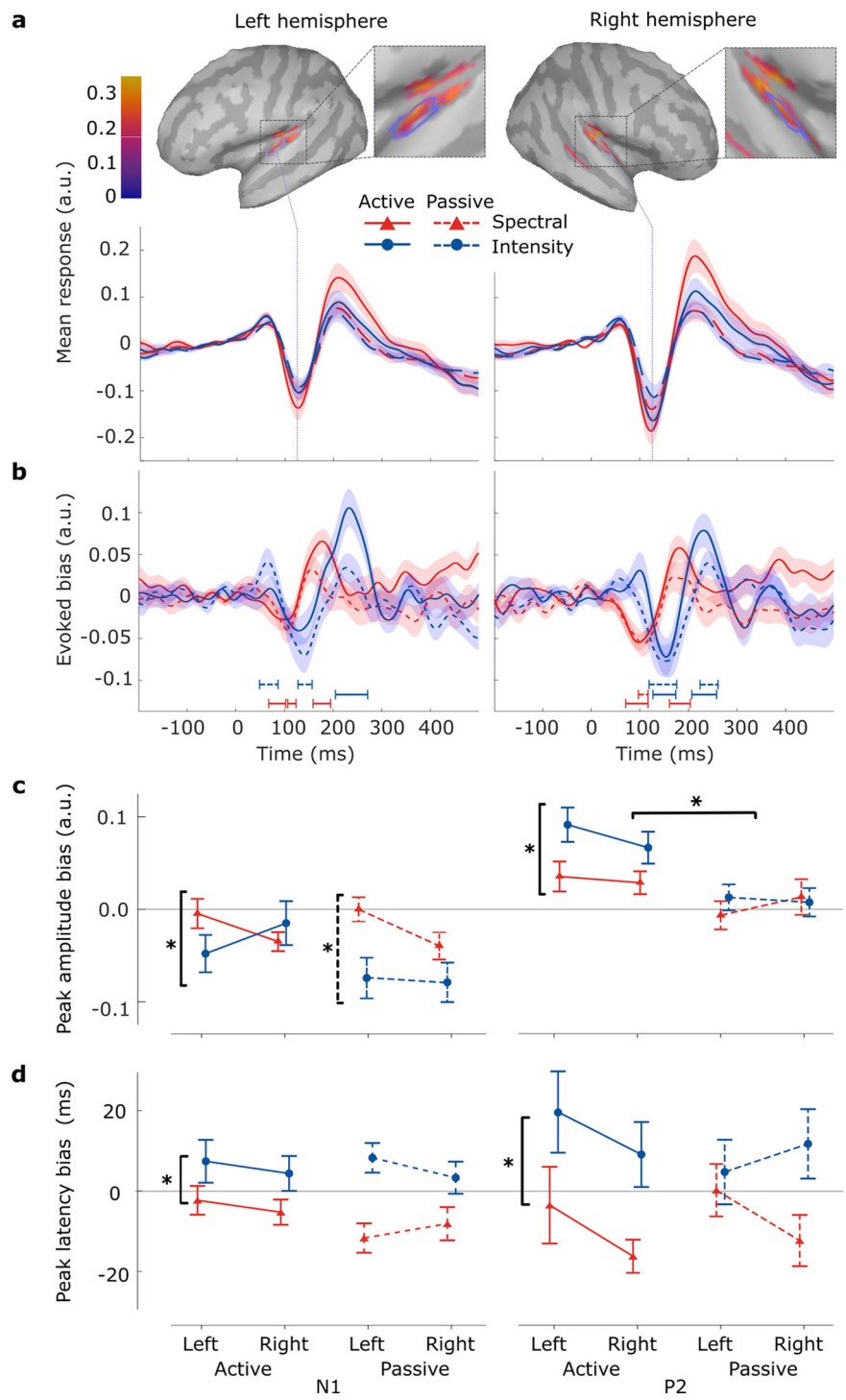

intensity than spectral cues (diff = 0.028 μV, $t(25) = 2.185$, $p = 0.038$, $d = 0.33$, 95% CI [0.01, 0.65]) and for active than passive listening (diff = 0.052 μV, $t(25) = 3.181$, $p = 0.004$, $d = 0.6$, 95% CI [0.19, 1.02]). Significant differences for component latencies were only found for cue type (Fig. 5d). For both N1 ($F(1, 25) = 10.99$, $p = 0.003$, $\eta_G^2 = 0.094$, 95% CI [0.00, 1.00]) and P2 ($F(1, 25) = 5.74$, $p = 0.024$, $\eta_G^2 = 0.046$, 95% CI [0.00, 1.00]), the spectral component appeared earlier than the intensity one (N1: diff = 0.013 s, $t(25) = 3.315$, $p = 0.003$, $d = 0.63$, 95% CI [0.25, 1.02]; P2: diff = 0.018 s, $t(25) = 2.396$, $p = 0.024$, $d = 0.43$, 95% CI [0.07, 0.79]). Taken together, attention mainly affected P2 amplitude biases and this effect appeared strongest for intensity cues. The bias again emerged pre-attentively, with a slight difference between hemispheres for the spectral cue type.

**Newborn listeners: looming bias elicited only by intensity cues**

After verifying the pre-attentive nature of the looming bias for both considered cues in the adult listener pool, we exposed 71 healthy full-term neonates in the deep sleep stage to the same stimuli. Apart from feasibility reasons[35,75,76], the deep sleep state ensured no attentive mechanisms were active.

In line with the procedure on our adult participants, we first performed a topographical analysis of the neural distribution at the scalp level. The cluster-based permutation test identified significant looming bias only for the intensity condition (clusterstat = $7.453*10^3$, $p = 0.006$, $d = 0.747$, 95% CI [0.434, 1.089]; Fig. 6d). Emerging at 270 ms after the change and initially lateralized to the right, the cluster subsequently moved more frontally,

**Fig. 6 | Change-evoked scalp potentials from newborns reveal auditory looming bias only for the intensity condition. a** Responses at the fronto-central electrode cluster for different cue types averaged across looming and receding trials. Shaded areas denote the standard errors of the means. **b** Looming versus receding neural responses for the change-locked intensity cue condition. The gray bar denotes the duration of the significant looming bias. **c** Looming versus receding neural responses for the change-locked spectral cue condition. **d** Topographic analysis of looming bias elicited by intensity cues. $N = 71$.

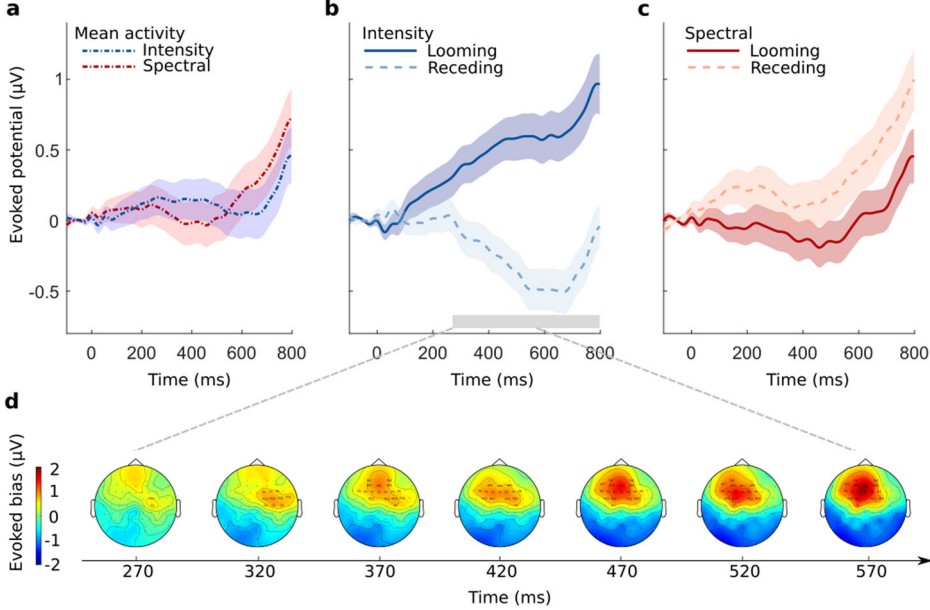

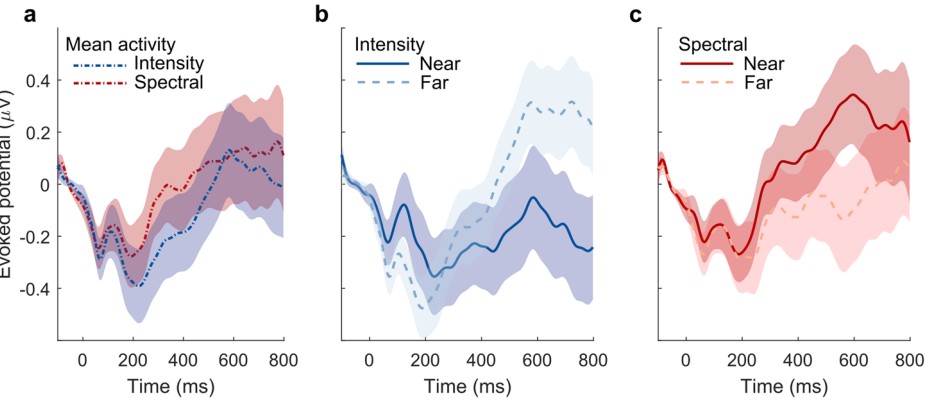

**Fig. 7 | Newborns' onset-evoked scalp potentials at the defined electrode cluster reveal no auditory position bias for any of the considered starting positions. a** Onset potentials averaged across trials of near and far positions. **b** Near versus far neural responses for the onset-locked intensity condition. **c** Near versus far neural responses for the onset-locked spectral condition. Shaded areas denote the standard errors of the means ($N = 71$).

finally solidifying in the frontocentral leads. The looming bias itself was found to intensify with elapsing time.

Based on the emerging topographical distribution, we extracted the average EEG time courses from an electrode cluster located in the fronto-central region of the scalp (see Newborn EEG analysis). The cluster activations averaged across looming and receding sounds appeared rather shallow until a rapid increase at around 400–500 ms after the event (Fig. 6a). The divergence between the looming and receding neural responses, representing the bias, depended on cue type (Fig. 6b, c). Consistent with the topographic analysis, the intensity looming bias first emerged 270 ms after the change event. Responses to looming and receding sounds drifted apart with progressing time, denoting a gradual intensification of the bias' amplitude (Fig. 6b). Contrary to that, neural looming and receding responses closely followed each other in the spectral condition, displaying no statistically significant evidence for difference in their time courses (Fig. 6c).

To test for position-specific effects, we additionally analyzed the event-related potentials locked to stimulus onset. Cluster-based permutation tests yielded no significant clusters. A $2 \times 2 \times 2$ Bayesian repeated-measures ANOVA with the factors cue type (spectral or intensity) and position (near or far) was performed. Bayes factor for exclusion (analysis of effects) for all factors, as well as their interaction, yielded no reliable evidence of a positional bias (cue type: $BF_{excl} = 4.142$; position: $BF_{excl} = 4.701$; cue type × position: $BF_{excl} = 4.851$). As for the adults, there was no credible evidence for

a difference between near and far sounds for either cue type (Fig. 7), indicating that the observed bias induced by intensity cues is specific to the change event.

Using template anatomical data for newborns and adjusted electrode locations, we inferred the generators of the recorded activity on the cortical surface (Fig. 8a). As for the adults, the change events evoked neural activity strongly focused on the posterior regions of the superior temporal gyri of both hemispheres, centered around the region of the HG. The change events also evoked activity in more distributed cortices of the newborns, including the superior and inferior temporal gyrus and occipital area. These observed activations might be attributed to object movement initiating rapid multi-sensory associative cortical processes, or the role of sleep in newborns' sensorimotor development[77]. We localized the HG bilaterally and extracted the corresponding cortical source responses.

Change-evoked neural source responses to looming versus receding stimuli were compared via cluster-based permutation statistics (Fig. 8b, c). Both cortices exhibited a response closely following the one found at the scalp level (Fig. 6b, c). Congruently, the HG time series in each hemisphere revealed a significant looming bias for the intensity condition, with the cluster appearing earlier for the left (230 ms, clusterstat = $8.46*10^3$, $p = 0.006$, $d = 3.721$, 95% CI [3.390, 4.084]) than for the right hemisphere (300 ms, clusterstat = $6.298*10^3$, $p = 0.018$, $d = 4.772$, 95% CI [4.364, 5.225]; Fig. 6b). In agreement with the scalp-level analysis, no statistically significant evidence of looming results were found for the spectral condition (Fig. 8c).

**Fig. 8 | Change-evoked HG activity in newborns reveals auditory looming bias only for the intensity condition. a** Brainmaps pooled across conditions and averaged within the time interval 250–300 ms. **b, c** Activity evoked by looming vs. receding sounds in left (left panels) and right (right panels) HG based on intensity (**b**) and spectral (**c**) cues. Gray areas denote the duration of significant temporal clusters. Shaded areas denote the standard errors of the means. $N = 71$.

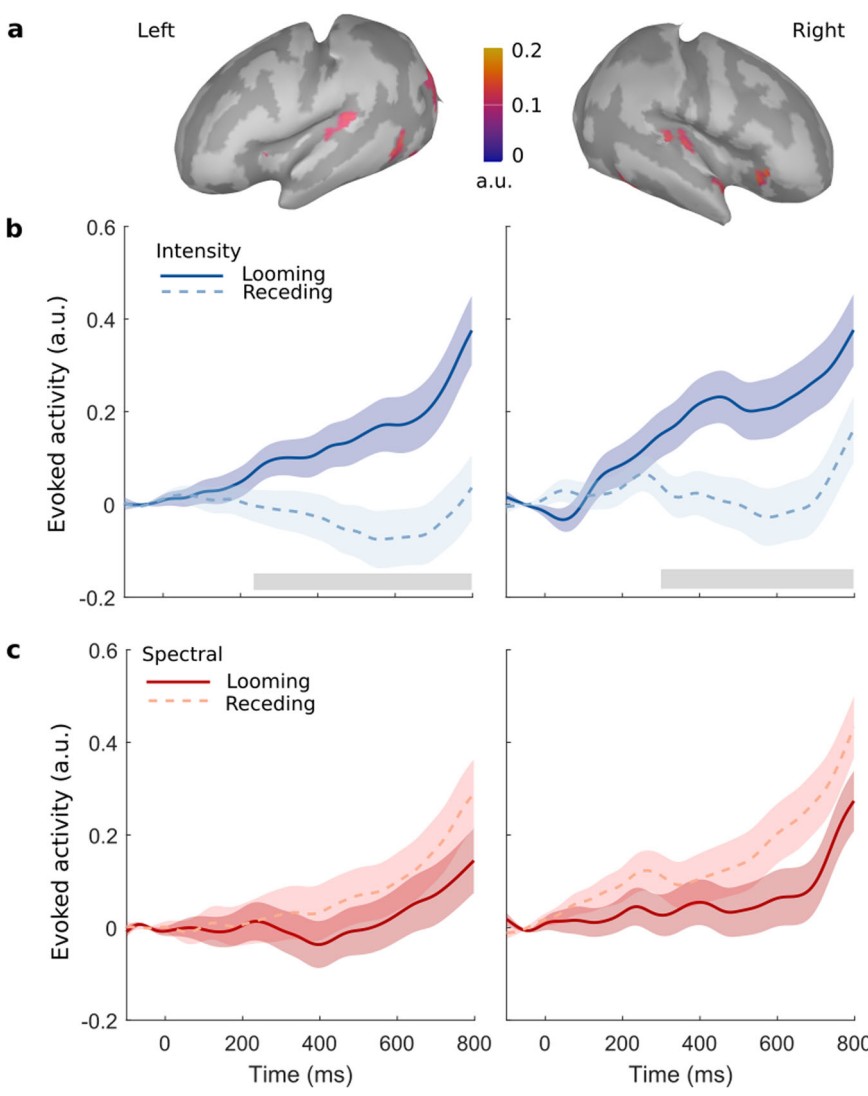

We further investigated the apparent lack of spectrally induced biases by applying Bayes factor hypothesis testing of evidence of absence [78]. A 2 × 2 repeated-measures ANOVA with the factors motion (looming or receding) and hemisphere (left or right) was performed. Bayes factor for exclusion (analysis of effects) yielded no credible evidence for a spectral bias, neither for the factor motion ($BF_{excl} = 3.15$) nor for its interaction with hemisphere ($BF_{excl} = 5.64$), corroborating the irrelevance of spectral looms to the HG of newborns.

## Discussion

In this study, we aimed to disentangle the inborn and learned aspects of the auditory looming bias. To this end, we analyzed the cortical responses from human adults and newborns to sounds perceptually moving along the distance dimension. We found the emergence of auditory looming bias in both age groups, yet it appeared to be processed differently depending on cue type. In adults, cue changes elicited neural biases in the HG as early as 50 ms after the change event and those were enhanced yet delayed for intensity compared to spectral cues. On the contrary, newborns demonstrated the bias only for intensity cues, beginning as early as 230 ms after the change event. This contrast between prenatally accessible intensity cues and postpartum changing spectral cues supports the idea that the looming bias comprises both innate and learned components.

## Related work

In adult participants, neural biases emerged stronger yet later in the intensity compared to the spectral condition. The relatively stronger responses align with the higher drift rates found behaviorally in the intensity condition. Evaluating spectral, and spatial cues is considered to require more complex processing [79] and create more subtle distance percepts [22,44]. Intensity cues should thus provide a more reliable perceptual read-out than spectral cues, especially under task-relevant conditions; the higher attentional modulation of neural biases induced by intensity cues is also consistent with this expectation. The relative delay of the intensity responses compared to the spectral ones may seem at first paradoxical, given that, behaviorally, intensity cues led to faster evidence accumulation. The reduced latency of neural biases in the spectral case may be due to mismatches in low-level auditory spatial tuning induced specifically by the transitions from native to flat spectral shapes. Thinking of this as a cue impoverishment connects well to previous investigations of spatial attention, which also found processing latencies, particularly around 50 ms, to be affected by impoverished auditory spatial cues [80].

Newborns exhibited the bias exclusively with the intensity cues. Both our analysis methods, cluster-based permutation, and Bayes factor hypothesis testing, provided no credible evidence for the presence of spectral looming bias in the newborn brain. In spite of the modifications of the sound characteristics taking place in utero during development, frequencies in the

range of 100–1000 Hz reach the fetus largely unchanged[25,81], enabling the processing of spectral acoustic information, at least for this low-frequency range. Yet the environment in utero, comprising liquid and essentially a low-pass filter for sounds, differs substantially from the one a newborn is postnatally exposed to. Especially given that our modifications affected frequencies beyond 1000 Hz, they have likely undergone essential distortion. Offering limited prenatal experience in the new environment, the absence of a spectral bias postnatally could align with the necessity of spatial associations for the spectral cues to be understood. Along these lines, early behavioral studies suggest that infants gradually acquire them during the first 18 months of their life[82,83].

We found early instances of auditory looming bias bilaterally at the level of the HG across attentional states and ages, while adults exhibited biases much earlier than newborns. This age-dependency by far exceeds expectations based on regular maturation speed-ups[84] and may suggest that adults establish more effective processes, specifically targeted towards detecting looming sounds. The particularly early biases in adults occurred more consistently for the right as compared to the left HG, an outcome potentially related to the right-hemispheric dominance of auditory spatial processing[85,86].

The HG lies on the superior surface of the temporal lobe and functionally houses the primary auditory cortex (Brodmann areas 41 and 42). As defined by the Desikan-Killiany atlas[67], where it is denoted as transverse temporal gyrus, it comprises the area between the rostral extent of the transverse temporal sulcus and the caudal portion of the insular cortex. The lateral fissure and the superior temporal gyrus are the medial and lateral boundaries, respectively[67]. The essential role of the auditory cortex emerges through previous work on the neural circuits of threat detection[3], suggesting that corticofugal projections from the auditory cortex to the inferior colliculus and lateral amygdala trigger defensive behavior. Silencing the auditory cortex in mice generally impedes auditory fear conditioning[87] and, in particular, their freeze and flight behavior in response to looming sounds[14]. Recordings in awake non-human primates also found neural populations within the primary[88] or secondary[13] auditory cortices to be biased in the same direction. Neuroimaging studies with human participants implicated regions such as the planum temporale and further uncovered widespread cortical networks that reflect the auditory looming bias[15–17]. Previous analyses suggest bottom-up directed connectivities from the primary auditory cortex to prefrontal areas[26], and our findings in sleeping newborns and inattentive adults further hint in this direction. Yet additional studies are needed to shed light on the nature and function of those networks beyond corticofugal projections.

## Limitations

Despite corroborative evidence, our findings should be conditional to cautious interpretation. The presence of intensity bias, the most salient cue type, in the newborn brain is a finding well in line with it having innate components potentially stemming from evolution. It is a possibility, though, that, to some degree, bias results from learning during intrauterine sensory development[89]. Although not found in our setting, the presence of spectral associations in the newborns' brains cannot be entirely refuted. The experimental set-up for the newborns inevitably differs methodologically from the one for the adults, potentially obscuring the result.

A possible reason for not finding an existing effect concerns the newborns' state of consciousness. We compared passively listening adults to sleeping newborns, a standard procedure in cross-age auditory research[35,75,76]. Although it has been demonstrated that awake and sleeping newborns show identical neural responses to sounds and changes in sound properties[90], the generalization of this to cue-specific looming sensation may come with some uncertainty. The response biases in adults were already diminished in the passive spectral condition, which might be a precursor for an even smaller effect in the corresponding newborn case, rendering its detection particularly challenging.

Another potential cause for the lack of finding biases induced by spectral cues concerns the sound characteristics. In previous newborn

studies, sounds were presented using loudspeakers in a sound-attenuating chamber. We presented the stimuli via headphones instead and simulated the acoustic transmission properties from a loudspeaker to the ear canal by individualized spectral filtering [91]. While for our adult listeners acoustical measurements were feasible to fully individualize, we had to rely on a partial individualization procedure based on anthropometric measurements[46] for the newborns. To further reduce the risk of insufficient HRTF individualization, we presented our stimuli from extremely lateral directions, where the HRTFs from the perceptually predominant ipsilateral side are among the least individual ones[92]. Nevertheless, there inevitably were inter-individual differences in the fit of the approximated HRTFs to the true ones. If newborns were sensitive to the spectral cues provided by their true HRTFs and were hindered solely due to insufficient HRTF individualization, the variance in the goodness of fit of the HRTFs should be reflected in the variance of the measured neural responses to the spectral condition. The variance we observed in the spectral condition is, however, comparable to the one emerging in the intensity one. Altogether, these considerations provide little evidence for acoustic inconsistencies being the underlying cause.

Another methodological downgrade for the newborn group concerns the use of template solutions for the inference of EEG source activity (brain anatomy and electrode locations). This lack of individualization degrades the EEG source localization accuracy[66] but does not affect the results on the scalp level. Since results were consistent across the scalp and source level, this methodological difference seems to play a minor role. Despite all taken measures and the more than twofold sample size of the newborn group, the possibility of them not sufficiently counterbalancing the imposed methodological limitations has to be acknowledged.

Across age groups, the use of EEG itself might have been a factor influencing the accuracy of our outcomes. EEG source localization relies on assumptions on the spread of activity, as the layers of bone and tissue between the cortical surface and the recording electrodes are inaccessible. As such, the process suffers from imprecision in the allocation of activity to its cortical generators. Due to its large inter-participant variability, auditory cortex localization is particularly difficult[18] in that respect. We made use of individual anatomical data and results from previous investigations[66] to infer activity from HG, attempting to limit such imprecisions to the most feasible degree. The analyzed inferred activity resembles the sought auditory cortex one, yet there could also be spill-over from secondary auditory regions. Future investigations with more fine-grained parcellations (e.g., TASH[93]) may give better insights into the dissociation of the two. Studies combining EEG with spatially more precise methods, such as fMRI and MEG, could, moreover, help better study the cortical generators involved in the bias. This study placed the target on the HG, aiming to investigate auditory cortical signatures of the looming bias; yet further whole-brain connectivity studies might aid in uncovering the larger network at play, including the multiple ROIs previously shown to be implicated in the biased perception of auditory looms.

The nature of auditory looms can be manifold. The implemented stimuli used in the present study comprise transition ramps in the order of 10 ms. The ecological validity of using a 10 ms duration to simulate looming or receding sounds depends on the natural soundscape and the types of events or objects being simulated. In certain real-world scenarios, such rapid changes may not be as common or not provide sufficient information for accurate perceptual judgments. In our experimental setting, however, keeping the transition phase short was not only crucial for good temporal isolation of neural processes but also to maintain consistency and comparability with highly relevant previous studies[8,26]. This ensured that the looming bias could be reliably elicited, particularly when utilizing the complementary cue type of spectral shapes.

Abrupt increases in sound intensity may also be judged as salient onset events[94] rather than motion events. While this confounds the interpretation of biases found for the intensity condition, it is less clear for the spectral condition. On the one hand, understanding the spectral cues is expected to rely on spatial associations[79,95]. Acquiring those associations is therefore

thought to facilitate the bias, purely from a spatial point of view and, by stimulus design, without intensity confounds. Nevertheless, associative learning could be possible, meaning that the significance of one cue gets learned purely based on an understanding of another. Although isolated in our study design, intensity, and spectral cues do not appear as such in nature, therefore obscuring the precise interdependencies. Associative learning may also explain why full motion cues are the most efficient in facilitating the warning mechanism of the looming bias[7]. To further investigate this question, it would be interesting to study the possibility of inducing looming bias with novel spectral cues. Those should have been acquired through directional localization training and with stimulus intensity being roved to rule out intensity associations[96].

## Conclusions

Taken together, we found that both human adults and newborns exhibit the auditory looming bias at the level of the HG during inattentive listening. The primary auditory cortex, a functional region within the HG, has previously been associated with the looming bias. Our results thus corroborate the notion that the auditory looming bias reflects an early, pre-attentive warning mechanism, potentially originating from activity within the primary auditory cortex.

However, the presence of this bias appears to be contingent on cue type, a finding consistent with the requirement for prior cue exposure. The auditory looming bias seems therefore to be partially innate, encoded through the evolutionary history of species, without the need for previous threat experience. Nevertheless, it remains flexible enough to effectively integrate new spatial cues acquired through lifelong exposure. How this cue universality is achieved remains to be elucidated.

## Data availability

Data are available at https://osf.io/4gdy2/[97].

## Code availability

Experimental paradigm and analysis scripts are available at https://osf.io/4gdy2/[97].

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

## Acknowledgements
We would like to thank Sophie Hanke, Tobias Greif, Regina Pfennigschmidt, Gábor P. Háden, and Eszter Rozgonyiné Lányi for their assistance during data collection, Martin Lindenbeck and David Meijer for their insights during data analysis, and Barbara G. Shinn-Cunningham and István Winkler for general advice. This research was funded by the Austrian Science Fund (FWF, I 4294-B and ZK66), the National Research Development and Innovation Office grant (NKFIH; ANN131305), and the Hungarian Academy of Sciences [Magyar Tudományos Akadémia (MTA)] through the János Bolyai grant (BO/00237/19/2). The funders had no role in the conceptualization, design, data collection, analysis, decision to publish, or preparation of the paper.

## Author contributions
R. Baum. and B.T. conceived the study. D.B. and R. Baum. designed the experiment, and K.I. and B.T. contributed to paradigm refinement. D.B. implemented and conducted the adult experiment with help from K.I. K.I. and D.B. curated the adult EEG dataset. B.T. and I.S. curated the newborn EEG and medical dataset. R. Baru., D.B., and R. Baum. analyzed the behavioral data. K.I. analyzed the neural data. K.I., R. Baum., R. Baru. and D.B. designed the data presentation and wrote the paper. K.I., R. Baum., B.T., and R. Baru. revised the paper. R. Baum., B.T., and I.S. acquired the funding, obtained ethical permissions, and managed the project.

## Competing interests
The authors declare no competing interests.
