## [Peer Review File · Communications Psychology]

23rd Nov 23

Dear Dr Baumgartner,

Thank you for your patience during the peer-review process. We apologize for the delay in reaching a decision. Your manuscript titled "Cortical signatures of auditory looming bias show exposure-based adaptation across the human lifespan" has now been seen by 3 reviewers, whose comments are appended below. You will see that they find your work of some potential interest. However, they have raised quite substantial concerns that must be addressed. In light of these comments, we cannot accept the manuscript for publication, but would be interested in considering a revised version that fully addresses these serious concerns.

We hope you will find the Reviewers' comments useful as you decide how to proceed. Should additional work allow you to address these criticisms, we would be happy to look at a substantially revised manuscript. If you choose to take up this option, please highlight all changes in the manuscript text file, and provide a detailed point-by-point reply to the reviewers.

Editorially, we consider it important that the revised manuscript includes a revised discussion and limitations section that does not overinterpret the findings, appropriately accounts for uncertainty in the findings (e.g., influence of prenatal exposure, influence of the intensity of the final sound heard), and fully notes the limitations of the study. Please attend to the reviewer's request for greater detail and clarification regarding the methods. We consider it important that you provide strong evidence for key claims via additional data analysis. Methodological concerns that cannot be addressed should be fully explicated in the limitations.

For manuscripts that report null results, we require the following:

- Evidence that the study is sufficiently powered to detect the smallest theoretically or pragmatically meaningful effect
- Bayes Factors or equivalence tests to interpret the null results
- Appropriate language to describe the results.

If the revision process takes significantly longer than five months, we will be happy to reconsider your paper at a later date, provided it still presents a significant contribution to the literature at that stage.

Please use the following link to submit your revised manuscript, point-by-point response to the Reviewers' comments with a list of your changes to the manuscript text (which should be in a separate document to any cover letter) and any completed checklist:

[link redacted]

Please do not hesitate to contact me if you have any questions or would like to discuss the required revisions further. Thank you for the opportunity to review your work.

Best regards,

Jennifer Bellingtier

on behalf of
Katherine Storrs, PhD
Editorial Board Member
Communications Psychology
orcid.org/0000-0001-9573-8654

EDITORIAL POLICIES AND FORMATTING

Editorial Policy: Policy requirements (Download the link to your computer as a PDF.)

Furthermore, please align your manuscript with our format requirements, which are summarized on the following checklist:

Communications Psychology formatting checklist

and also in our style and formatting guide Communications Psychology formatting guide .

* TRANSPARENT PEER REVIEW: Communications Psychology uses a transparent peer review system. This means that we publish the editorial decision letters including Reviewers' comments to the authors and the author rebuttal letters online as a supplementary peer review file. However, on author request, confidential information and data can be removed from the published reviewer reports and rebuttal letters prior to publication. If your manuscript has been previously reviewed at another journal, those Reviewers' comments would not form part of the published peer review file.

* CODE AVAILABILITY: All Communications Psychology manuscripts must include a section titled "Code Availability" at the end of the methods section. In the event of publication, we require that the custom analysis code supporting your conclusions is made available in a publicly accessible repository; please choose a repository that provides a DOI for the code; the link to the repository

and the DOI must be included in the Code Availability statement. Publication as Supplementary Information will not suffice. We ask you to prepare and upload code at this stage, to avoid delays later on in the process.

*** DATA AVAILABILITY:**

All Communications Psychology research manuscripts must include a section titled "Data Availability" at the end of the Methods section or main text (if no Methods). More information on this policy, is available at <http://www.nature.com/authors/policies/data/data-availability-statements-data-citations.pdf>.

At a minimum the Data availability statement must explain how the data can be obtained and whether there are any restrictions on data sharing. Communications Psychology strongly endorses open sharing of data. If you do make your data openly available, please include in the statement:

We recommend submitting the data to discipline-specific, community-recognized repositories, where possible and a list of recommended repositories is provided at <http://www.nature.com/sdata/policies/repositories>.

If a community resource is unavailable, data can be submitted to generalist repositories such as figshare or Dryad Digital Repository. Please provide a unique identifier for the data (for example a DOI or a permanent URL) in the data availability statement, if possible. If the repository does not provide identifiers, we encourage authors to supply the search terms that will return the data. For data that have been obtained from publicly available sources, please provide a URL and the specific data product name in the data availability statement. Data with a DOI should be further cited in the methods reference section.

REVIEWER EXPERTISE:

Reviewer #1 neurological auditory development

Reviewer #2 developmental perception/neuroscience

Reviewer #3 audition

Reviewer #1 (Remarks to the Author):

The paper reports on auditory experiments with human adults and newborns on the phenomenon of auditory looming bias (i.e., the fact that approaching sounds are more strongly reacted to than receding sounds). The bias can be perceived thanks to different auditory cues, two of which are investigated in the paper: sound intensity, and spectral shape. While adults show the looming bias with both intensity and spectral cues, newborns show it only for intensity cues.

I find the study interesting and novel, especially with regards to the comparison between the two

developmental stages. I appreciate the behavioral part of the adult experiment where looming bias was confirmed to occur without attention, thereby better motivating the newborn data collected during sleep, and the use of Bayesian statistics to confirm the lack of spectrally induced looming bias in newborns. I believe that the manuscript would benefit from addressing the following issues:

(1) **PRENATAL AUDITORY EXPERIENCE.** The paper aims at “disentangle[ing] the inborn and learned aspects of the auditory looming bias”. While achieving this goal with an experiment with newborns is the obvious approach, I miss the authors acknowledging the limitation of not accounting for prenatal auditory experience. In my view, throughout the paper it is not elaborated upon enough that and which type of auditory stimulation is available in utero. The Heschl’s gyrus, hosting the primary auditory functionality develops already around the 24 weeks of gestation and fetal hearing is functional before birth. During development, sounds passing through the mother’s abdomen and amniotic fluid undergo spectral modification and attenuation but frequencies between 100 Hz and 1,000 Hz reach the fetus largely unchanged. Please see Chládková and Paillereau, 2020 and Granier-Deferre et al., 2011 and references therein for overviews. On page 3, the authors do mention which types of auditory cues are prenatally accessible, but to substantiate their claim they refer to only one study, that is quite dated (Querleu et al., 1988). They claim that intensity cues are, and spectral cues are not prenatally accessible. Please elaborate whether this is true for moving sounds only, or is it a general characteristic of prenatal language environment. There is some evidence that fetuses process spectral information (Granier-Deferre et al., 2011) and I believe that the authors should integrate it in the Discussion and/or Introduction.

(2) **PAC.** Throughout the paper, the cortical region that is being shown to be associated with the looming bias is referred to as the primary auditory cortex (PAC). PAC is a functional specification, and not an anatomical one and using it almost exclusively throughout the paper might in my opinion not be 100% correct, see for example Zoellner et al., (2019) for a discussion on PAC localization issues. Especially given the fact that based on the supplied reconstructed activity maps, the activated regions seem to encompass more than only the first transverse temporal gyrus (i.e., Heschl’s gyrus) that has been typically associated with primary auditory activity. Is it a spill-over effect or is the ROI defined so broadly? On a related note, the authors specify that for the extraction of the evoked PAC activity they have used the “transverse temporal region” label from the Desikan-Killiany atlas in Freesurfer. This atlas lacks some important detail with respect of the auditory regions and I wonder why the authors did not use the more detailed Destrieux (2010) atlas, or segmentation tools specifically designed for delineating the auditory cortex regions (Dalboni da Rocha et al., 2020)? A more general question about the localization analysis concerns the issue of: if not in the areas associated with primary auditory processing, then where could the looming bias be observed? Please motivate further and discuss the importance of the finding. Please also note that from the introduction, it seems that it was not only the primary auditory region that was investigated, but that this region’s activity was related to the looming bias the strongest (i.e., the authors do not explicitly state that they performed an ROI analysis). From Figures 4 and 6, it seems that the activation was indeed observed in other regions as well, but this fact has not been discussed. Please provide clarifications, beyond the one sentence in the Methods section (line 615), to avoid confusion.

(3) **LINEAR BALLISTIC ACCUMULATOR MODEL.** The analysis of the behavioral data is quite sophisticated and impressive. However, I wonder what its added value is over simpler solutions (e.g., linear mixed effects models). Could the reported results be replicated with LME? Why is it the most appropriate way of analyzing this type of data. Without experience with this type of approaches, I personally find it hard to judge the robustness of the reported results, and the appropriateness of the analytical choice (please pardon my ignorance), especially given that the EEG data were analyzed

with a more traditional approaches. Please justify in more depth the analytical choices.

Minor Points:

- I would not refer to samples of N = 35 and N = 71 as large (line 87)
- It would be informative for the readers to have access to example stimuli on the OSF site of the study
- The authors refer to “supplementary sec. [A,B,C]” in the manuscript but the sections are called “Appendix [A,B,C]” in my materials. Please be consistent
- Please spell out and explain “HRTF” upon first mention in the text
- In the analysis of newborn data, the authors report the “cluster activations averaged across looming and receding sounds showed a slow increase followed by a drop in activity, and a more rapid subsequent increase” (line 225). Why don't the authors report such averaged activations across conditions for adult participants as well?
- It is not necessary to explain the notation of “(mean \pm standard deviation)” more than once in the text
- For processing software described in ‘Additional review materials, Reporting Summary’, please specify the Freesurfer version used for MRI segmentation. I believe that for completeness, Sequence and imaging parameters should be reported, together with a specification of whole brain area of acquisition

References cited:

- Chládková, K., Paillereau, N., 2020. The What and When of Universal Perception: A Review of Early Speech Sound Acquisition. *Lang Learn* 70, 1136–1182. <https://doi.org/10.1111/lang.12422>
- Dalboni da Rocha, J.L., Schneider, P., Benner, J., Santoro, R., Atanasova, T., Van De Ville, D., Golestani, N., 2020. TASH: Toolbox for the Automated Segmentation of Heschl's gyrus. *Sci Rep* 10, 3887. <https://doi.org/10.1038/s41598-020-60609-y>
- Destrieux, C., Fischl, B., Dale, A., Halgren, E., 2010. Automatic parcellation of human cortical gyri and sulci using standard anatomical nomenclature. *Neuroimage* 53, 1–15. <https://doi.org/10.1016/j.neuroimage.2010.06.010>
- Granier-Deferre, C., Ribeiro, A., Jacquet, A.Y., Bassereau, S., 2011. Near-term fetuses process temporal features of speech. *Dev Sci* 14, 336–352. <https://doi.org/10.1111/j.1467-7687.2010.00978.x>
- Querleu, D., Renard, X., Versyp, F., Paris-Delrue, L., Crèpin, G., 1988. Fetal hearing. *European Journal of Obstetrics and Gynecology and Reproductive Biology* 28, 191–212. [https://doi.org/10.1016/0028-2243\(88\)90030-5](https://doi.org/10.1016/0028-2243(88)90030-5)
- Zoellner, S., Benner, J., Zeidler, B., Seither-Preisler, A., Christiner, M., Seitz, A., Goebel, R., Heinecke, A., Wengenroth, M., Blatow, M., Schneider, P., 2019. Reduced cortical thickness in Heschl's gyrus as an in vivo marker for human primary auditory cortex. *Hum Brain Mapp* 40, 1139–1154. <https://doi.org/10.1002/hbm.24434>

Reviewer #2 (Remarks to the Author):

Review of COMMSPSYCHOL-23-0286-T by Baumgartner and colleagues

The authors report an interesting and valuable study on a difficult participant group (newborn

infants), purportedly demonstrating the presence of a bias in processing for auditory looming (over receding) stimuli in adults and newborns in the intensity change domain, but and a bias for looming in adults but not newborns in the spectral domain.

Overall I am positive about this work. However, I have some reservations about the claims the authors make and the level of control they have in the experiments. I cannot recommend publication in the article's current form, not least because of apparent mistakes in reporting.

Rationale and interpretation:

The authors indicate that their aim is to disentangle inborn vs learned responses to auditory looming. To do this they present changes in "distance" by means of changes of intensity (accessible to fetuses prenatally) and spectral shape (not accessible prenatally). In newborns the authors find a looming bias only when using intensity and therefore claim the looming bias has an innate component ("the bias comprises innate components while flexibly incorporating the spatial cues acquired through lifelong exposure"). Because intensity info is accessible prenatally I am not convinced that we can rule out a role for prenatal experience in addition to or instead of phylogenetic factors. This is perhaps less likely given the comparative literature that the authors cite, but I think that this should at least be considered as a possibility. The authors also seem rather fixated on the view of looming biases being related to threat. Perhaps this is because of their phylogenetic stance. However, there is the possibility that differences between looming and receding can be learned via non-threatening but still salient multisensory information both pre and postnatally.

Design decisions:

Somewhat more problematic for interpretation of the newborn findings, I am not convinced that the authors' effects are due to changes in intensity. In the looming and receding conditions, changes in intensity from soft to loud (looming) vs loud to soft (receding) are perfectly confounded with the intensity of the last stimulus presented (loud for looming and soft for receding). Unless I have missed a clever feature of design or analysis which controls for this, I am concerned that the newborn findings could be due just to the intensity of the last stimulus presented and its continued presence. Perhaps this might explain what the authors do not seem to pick up very defined ERPs from the newborn group (making it difficult for them to undertake the comparisons in peaks employed with adults). Related to this question, it is worth asking whether newborns can notice a 5dB change. Do the authors have evidence to corroborate this design choice?

There is also a more subtle matter of interpretation. The intensity change happens within only a 10 ms window. The authors state that the transition phase itself was kept very short (10ms) to assure high temporal precision in the analysis of neural responses evoked by the change event. To what extent is this information really relevant to picking up looming vs receding of real objects in the real world?

Some other questions about rationale, design, and interpretation:

In adults the authors find larger results in the active vs passive condition (i.e. when attention is at play). Then why having the newborns in deep sleep vs e.g. awake or active sleep?

In the discussion, the authors suggest that spatial associations are required for spectral looming cues to be learned. And yet, it seems reasonable that intensity associations would be enough given the correlations available. Can they clarify this case?

The authors don't explain why they identified a particular sample size or report any power analysis. How was this matter approached?

There is little in the discussion which qualifies the claim that spectral information is not used by newborns. Given that individual subject HTRFs are not measurable from newborns, this is a limitation which could have led to the null effect compared to the adults. The authors say something about similar variances between spectral and intensity responses, but do not properly explain this.

It's unclear to me why the authors conclude that they newborn source localisation was sufficiently accurate given they replicated the same effects across both PAC and STS (albeit with smaller effects in STS). Was there a significant difference in amplitudes between STS and PAC? This seems particularly important given that the authors argue that the looming bias is present in PAC in both adults and infants.

Results reporting:

In Figure 3 and the related text, there appear to be inconsistencies and errors. The figure (3C) indicates a difference in latency between spectral and intensity, whereas the text suggests a difference between active and passive (lines 159-160). Why does this difference in latency NOT show up in Fig 3A? There seems to be a problem here. It is not entirely clear what Figure 3D is showing. The y axis suggests this is peaks, but then this seems to be difference waves. This is rather unintuitive and confusing.

in the adults sample, why are ERPs calculated only on 1 electrode (Cz) as opposed to a cluster (conversely, in newborns they use a fairly large cluster)? This seems a strange decision. Also when have the authors chose to compare peak amplitudes rather than, for instance, mean individual amplitudes given problems with peak amplitudes as a measure?

Lastly, I feel the authors could do more to clarify the nature of the drift rate latent variable which they extracted from the adult behavioural data.

Detailed comments:

The authors, in their EEG analysis claim to be investigating looming biases in "sensor space", but then proceed to an analysis of just the vertex electrode. Sensor space seems like an incorrect label for this kind of analysis. Just say "In EEG gathered from the scalp"?

Change "no significance" to "no significant difference in xx" throughout.

The authors introduced static sounds to ensure that ppts could not predict the stimulus category from the start, however they might have done purely based on the intensity.

Artifact rejection: excluding channels with artefacts with a criterion of -200:800 microvolt seems excessively lenient for adult data I think?

Filtering: .5 is a fairly high high pass, especially for adults (and in fact their infants' high pass is much lower)

in newborns "Trials numbers were equalised across conditions within each subject by removing trials

equally distributed across the recordings in order to match the minimum amount within the subject". This seems an odd practice given the robustness of parametric analysis to small violations of assumptions. How did the authors decide which trials to remove?

line 443: newborns were presented with 50 trials per condition vs line 653 average of 82 valid trials per condition.

Reviewer #3 (Remarks to the Author):

Main points

- This is an extensive, well-conducted study that presents an impressive amount of work. The large sample sizes, refined source analysis procedure, and advanced statistical methods are particularly noteworthy.
- In theory, the idea to contrast spectral and intensity cues to distinguish between the learned and innate components of the looming bias, respectively, is indeed compelling. However, even the adult EEG data only showed a relatively small bias in response to spectral cues in the passive condition. My main concern is that the absence of this effect in the newborns cannot simply be attributed to their inability to use these cues, as the newborn experiment inevitably represents a downgrade in almost every methodological regard. For instance, no MRIs and electrode positions could be obtained, the stimuli were template-based rather than individualised, and the newborns were sleeping rather than just inattentive. Hence, it is not entirely clear if there was indeed a difference between groups in that regard or if the data just weren't precise enough to uncover an effect in the newborns. Although some limiting factors are discussed, the authors are trying a bit too hard to sell their results at face value and I would welcome a somewhat more cautious interpretation of the results.
- Although quite some effort has been taken to conduct the source analyses (individual MRIs and channel positions in adult sample) and the use of the dSPM algorithm resulted in very fine-grained distributed source reconstructions, the discussion of the cortical generators involved remained very superficial and their anatomical description is mostly inadequate. Please see comments re Lines 183, 186, 241, and 615 below. The authors should update the relevant sections by correctly naming the different areas that showed activity and describe which processing steps they are commonly associated with, particularly with respect to PAC and planum temporale.

Specific comments:

- Abstract: This needs a little more content. What exactly did the source localisations reveal in your EEG data?
- Line 68: Too vague, please describe these findings briefly.
- Line 109: This is probably too short to even notice the jitter.
- Fig. 1: The lower panel of B is hard to grasp and requires more explanation. For the three signals with a non-flat spectral envelope, it looks as if you just altered the overall power, i.e., DC offset.
- Line 132: Please back this up with some stats.
- Line 178: I don't see what this additional analysis adds to the manuscript. You are merely measuring an unspecific sound onset response here (e.g., Krumbholz et al. 2003, Cerebral Cortex), so

it is not surprising that there were no differences between conditions.

- Line 183: You extracted source waveforms from entire Heschl's gyrus, so this title doesn't apply. Please see comment Line 615.
- Line 186: This is a gross misdescription of the results. Activity was evident around PAC and in planum temporale, rather than the neighbouring STG, and leaked into the insula. The latter effect is typical for EEG source reconstructions and demonstrates their imprecision.
- Line 193: These waveforms look a lot noisier than the scalp ERPs and the pattern in the results is also less clear cut. It seems too much a simplification to just state that the bias elicited by spectral cues has a shorter latency.
- Line 226: To me these ERPs look flat until about 500 ms after the acoustic change.
- Line 241: There is quite a bit of activity outside auditory cortex that should be mentioned too.
- Line 485: This raises the possibility that spectral cues were per se more difficult to detect for newborns as a template HRTF was used.
- Line 615: PAC is not the same as the transverse temporal gyrus, i.e., Heschl's. The medial part is generally considered PAC, while the anterolateral part contains secondary areas.
- Line 620: Using the grand mean across subjects and then specifying time windows based on this would have been a more convincing approach.
- Line 641: The high-pass cutoff at 0.05 Hz is much lower than the one for the adults (0.5 Hz). There appear to be strong drifts in the newborn ERPs. Could this be due to the low cutoff?
- Line 670: Why this time window?

Kurt Steinmetzger

Point-by-point response to the Reviewers' comments

General note: Line numbers in our responses refer to the annotated manuscript.

REVIEWER EXPERTISE:

Reviewer #1 neurological auditory development

Reviewer #2 developmental perception/neuroscience

Reviewer #3 audition

REVIEWER #1 (REMARKS TO THE AUTHOR):

The paper reports on auditory experiments with human adults and newborns on the phenomenon of auditory looming bias (i.e., the fact that approaching sounds are more strongly reacted to than receding sounds). The bias can be perceived thanks to different auditory cues, two of which are investigated in the paper: sound intensity, and spectral shape. While adults show the looming bias with both intensity and spectral cues, newborns show it only for intensity cues. I find the study interesting and novel, especially with regards to the comparison between the two developmental stages. I appreciate the behavioral part of the adult experiment where looming bias was confirmed to occur without attention, thereby better motivating the newborn data collected during sleep, and the use of Bayesian statistics to confirm the lack of spectrally induced looming bias in newborns. I believe that the manuscript would benefit from addressing the following issues:

1. **PRENATAL AUDITORY EXPERIENCE.** The paper aims at “disentangle[ing] the inborn and learned aspects of the auditory looming bias”. While achieving this goal with an experiment with newborns is the obvious approach, I miss the authors acknowledging the limitation of not accounting for prenatal auditory experience. In my view, throughout the paper it is not elaborated upon enough that and which type of auditory stimulation is available in utero. The Heschl’s gyrus, hosting the primary auditory functionality develops already around the 24 weeks of gestation and fetal hearing is functional before birth. During development, sounds passing through the mother’s abdomen and amniotic fluid undergo spectral modification and attenuation but frequencies between 100 Hz and 1,000 Hz reach the fetus largely unchanged. Please see Chládková and Paillereau, 2020 and Granier-Deferre et al., 2011 and references therein for overviews. On page 3, the authors do mention which types of auditory cues are prenatally accessible, but to substantiate their claim they refer to only one study, that is quite dated (Querleu et al., 1988). They claim that intensity cues are, and spectral cues are not prenatally accessible. Please elaborate whether this is true for moving sounds only, or is it a general characteristic of prenatal language environment. There is some evidence that fetuses process spectral information (Granier-Deferre et al., 2011) and I believe that the authors should integrate it in the Discussion and/or Introduction.

Response:

We thank the reviewer for their elaborate feedback and information on prenatal sound processing. We reworked the manuscript to include the requested – and very necessary – information:

Introduction (lines 87-94):

“HG is already developed around the 24th week of gestation and fetal hearing is functional before birth. Sounds, passing through the mother’s abdomen and amniotic fluid during development, undergo spectral modification and attenuation. Intensity ramps are already prenatally accessible [22] and evidence suggests that spectral information is also processed [23]. Yet newborns are additionally subject to abrupt changes in the environment postpartum. This substantially affects the characteristics of spectral cues, thereby necessitating a new acquisition or adaptation process.”

Discussion (lines 368-397):

“In spite of the modifications of the sound characteristics taking place in utero during development, frequencies in the range of 100 - 1000 Hz reach the fetus largely unchanged [23, 40], enabling the processing of spectral acoustic information at least for this low-frequency range. Yet the environment in utero, comprising liquid and essentially a low-pass filter for sounds, differs substantially from the one a newborn is postnatally exposed to. Especially given that our modifications affected frequencies beyond 1000 Hz, they have likely undergone essential distortion. Offering limited prenatal experience in the

new environment, the absence of a spectral bias postnatally could align with the necessity of spatial associations for the spectral cues to be understood. Along these lines, early behavioral studies suggest that infants gradually acquire them during the first 18 months of their life [41, 42].”

2. PAC. Throughout the paper, the cortical region that is being shown to be associated with the looming bias is referred to as the primary auditory cortex (PAC). PAC is a functional specification, and not an anatomical one and using it almost exclusively throughout the paper might in my opinion not be 100% correct, see for example Zoellner et al., (2019) for a discussion on PAC localization issues. Especially given the fact that based on the supplied reconstructed activity maps, the activated regions seem to encompass more than only the first transverse temporal gyrus (i.e., Heschl’s gyrus) that has been typically associated with primary auditory activity. Is it a spill-over effect or is the ROI defined so broadly?

Response:

We thank the reviewer for highlighting this confusing point. We followed the mentioned guidelines and proceeded by:

- Addition to the introduction, to better describe and dissociate the regions (lines 73-77): “Apart from that, localization of the auditory cortex in neuroimaging studies is non-trivial: although the medial part of the anatomical region of Heschl’s gyrus (HG) is generally considered to host the primary auditory cortex, it remains a functional definition suffering large inter-subject variability [18].”
- Removing the functional characterization of “PAC” from the results section and only referring to the actual anatomical regions in question, namely Heschl’s gyrus (HG) throughout the results section.
- Adding, in Figure 4, a magnified figure of the brainmaps in question, to clearly show the defined regions.
- Better describing the activations visible on said brainmaps (lines 235-239): “Both the left and right HG exhibited stereotypical auditory evoked responses for all considered conditions. In addition, we found high activity at more posterior regions (planum temporale), while activations seem to have leaked into the posterior regions of the insular cortex. Further investigation of these ROIs outside HG was out of scope of the current study.”
- Addition to the discussion (lines 406-411): “The HG lies on the superior surface of the temporal lobe and functionally houses the primary auditory cortex (Brodmann areas 41 and 42). As defined by the Desikan-Killiany atlas, where it is denoted as transverse temporal gyrus, it comprises the area between the rostral extent of the transverse temporal sulcus and the caudal portion of the insular cortex. The lateral fissure and the superior temporal gyrus are the medial and lateral boundaries, respectively [49].”
- Addition to the discussion (lines 471-486): “EEG source localization relies on assumptions on the spread of activity, as the layers of bone and tissue between the cortical surface and the recording electrodes are inaccessible. As such, the process suffers from imprecisions in the allocation of activity to its cortical generators. Due to its large inter-subject variability, auditory cortex localization is particularly difficult [18] in that respect. We made use of individual anatomical data and results from previous investigations [31] to infer activity from HG, attempting to limit such imprecisions to the most feasible degree. The analyzed inferred activity resembles the sought auditory cortex one, yet there could also be spill-over from secondary auditory regions. Future investigations with more fine-grained parcellations (e.g. TASH [53]) may give better insights on the dissociation of the two. Studies combining EEG with spatially more precise methods, such as MEG, could, moreover, help better study the cortical generators involved in the bias. This study placed the target on the HG to investigate auditory cortical signatures of the looming bias; yet further whole-brain connectivity studies might aid towards uncovering its functional role and the larger network at play, including the multiple ROIs previously shown to be implicated in the biased perception of auditory looms.”

3. On a related note, the authors specify that for the extraction of the evoked PAC activity they have used the “transverse temporal region” label from the Desikan-Killiany atlas in Freesurfer. This atlas lacks some important detail with respect of the auditory regions and I wonder why the authors did not use the more detailed Destrieux (2010) atlas, or segmentation tools specifically designed for delineating the auditory cortex regions (Dalboni da Rocha et al., 2020)?

Response:

We thank the reviewer for pointing out this lack of clarification. We chose the Desikan-Killiany parcellation mainly for consistency to previous work (Bidelman & Meyers, 2020; Ignatiadis et al., 2021). Moreover, we considered the Destrieux atlas to be too fine grained for the spatial localization accuracy practically achievable through EEG, even though we used high-density EEG, individualized brain anatomies and electrode positions, and tested different inverse solutions beforehand (Ignatiadis et al., 2022). In comparison, the region in question is very similar between the two atlases, although the Destrieux parcellation distinguishes between the transversetemporal gyrus and sulcus, which Desikan-Killiany does not:

Destrieux: 26 vertices, 5.75 cm² area (gyrus, blue) Desikan-Killiany: 34 vertices, 6.16 cm²
40 vertices, 7.04 cm² area (all incl. sulcus, red)

Hence, we do not think that these subtle differences would change our results given the limitations in localization accuracy we face with EEG.

Our motivation and implementation is now better clarified in the methods section (lines 820-823):

“For consistency and comparability with previous relevant literature [24, 25], evoked HG activity was extracted according to the Desikan-Killiany parcellation scheme as defined in Brainstorm (transverse temporal region) [49]”

Regarding the TASH toolbox (Dalboni da Rocha et al., 2020), we would like to thank the reviewer for bringing it to our attention. We will certainly consider it for our upcoming research targets. For the present study, we are more cautious now about the interpretation of our findings as originating from the primary, as opposed to secondary, auditory cortices.

4. A more general question about the localization analysis concerns the issue of: if not in the areas associated with primary auditory processing, then where could the looming bias be observed? Please motivate further and discuss the importance of the finding. Please also note that from the introduction, it seems that it was not only the primary auditory region that was investigated, but that this region's activity was related to the looming bias the strongest (i.e., the authors do not explicitly state that they performed an ROI analysis). From Figures 4 and 6, it seems that the activation was indeed observed in other regions as well, but this fact has not been discussed. Please provide clarifications, beyond the one sentence in the Methods section (line 615), to avoid confusion.

Response:

As seen by the brainmaps, activity was not just found on the transversetemporal gyrus, but this is the region we targeted because animal studies suggest it to play a crucial role, while previous literature on humans also hinted to its importance (Bidelman & Myers, 2020; Ignatiadis et al., 2021). We agree that this should be better clarified within the introduction section, and did so in lines 109-111:

“Investigations were done at the level of the scalp as well as HG; a choice made based on prior literature [24,25] and due to it comprising the functionality of the primary auditory cortex”

For this comparative study, we targeted the early cortical processing taking place in HG. Identifying the role of this region as part of the entire cortical network would be an interesting target of follow-up investigations. This is now specified in the discussion in line 483-486:

“This study placed the target on the HG, aiming to investigate auditory cortical signatures of the looming bias; yet further whole-brain connectivity studies might aid towards uncovering the larger network at play, including the multiple ROIs shown to be implicated in the biased perception of auditory looms.”

5. LINEAR BALLISTIC ACCUMULATOR MODEL. The analysis of the behavioral data is quite sophisticated and impressive. However, I wonder what its added value is over simpler solutions (e.g., linear mixed effects models). Could the reported results be replicated with LME? Why is it the most appropriate way of analyzing this type of data. Without experience with this type of approaches, I personally find it hard to judge the robustness of the reported results, and the appropriateness of the analytical choice (please pardon my ignorance), especially given that the EEG data were analyzed with a more traditional approaches. Please justify in more depth the analytical choices.

Response:

We thank the reviewer for pointing out this lack of clarity. We now expanded the description in the results section as follows (lines 167-178):

“We selected this model-based approach because of its advantage in accounting for the speed-accuracy trade-off on a trial-by-trial level [28] as well as the different uncertainty levels across subjects [27]. In this modeling framework, an evidence accumulation process is started for every choice option and trial; the accumulator hitting the response threshold first decides the choice as well as the response time. To study the presence of looming bias, our latent variable of interest was the drift rate, which quantifies the velocity of evidence accumulation towards a response in a forced choice task [29]. With drift rates fitted for every stimulus condition, simulated data demonstrated high agreement with the actual hit rates and response times across all conditions ($p > 0.33$; Fig. 2A).”

As the reviewer was wondering about the robustness of the reported results, we re-ran the fitting procedure with the double amount of samples (8000 instead of 4000) and obtained very similar results. This further improved the convergence of the MCMC (Gelman-Rubin diagnostic decreased from 1.012 to 1.006; best possible score is 1).

Analyzing the behavioral data with standard LME approaches introduces several disadvantages. First, analyses are performed separately for response times and hit rates and thus do not account for their trade-off. Second, both data types are typically quite skewed in their distributions and thus residuals often do not turn out as normally distributed as assumed by LME models. Third, standard LME models do not allow to account for the large inter-individual differences in response accuracy (i.e., substantially different amounts of trials with correct responses) we observed. That said, we performed this analysis based on LME modeling. It also reveals looming biases on a behavioral level in terms of higher accuracies or lower response times depending on cue type. Biases in accuracy are found significant only for spectral cues [$b = -0.14$, $SE = 0.03$, $t(27) = -4.23$, $p < .001$] and biases in response times are found only for intensity cues [$b = 0.03$, $SE = 0.01$, $t(26.3) = 2.71$, $p = .012$]. We opted against reporting this additional analysis in the manuscript because it suffers from the disadvantages stated above and does not provide additional insight.

Minor Points:

6. I would not refer to samples of $N = 35$ and $N = 71$ as large (line 87)

Response:

Agreed; “large” was removed from the sentence. (line 100)

7. It would be informative for the readers to have access to example stimuli on the OSF site of the study

Response:

Agreed. Examples were added as part of the experimental scripts on the OSF site.

8. The authors refer to “supplementary sec. [A,B,C]” in the manuscript but the sections are called “Appendix [A,B,C]” in my materials. Please be consistent

Response:

We thank the reviewer for pointing out this inconsistency. “Supplementary” was corrected to “Appendix” throughout the manuscript.

9. Please spell out and explain “HRTF” upon first mention in the text

Response:

Apologies for this omission; HRTF is now spelled out and explained upon first mention, in the introduction (lines 103-105):

“Stimuli were filtered by sets of head-related transfer functions (HRTFs), namely a set of filters representing the sound modifications induced by one’s pinna, head and torso.”

10. In the analysis of newborn data, the authors report the “cluster activations averaged across looming and receding sounds showed a slow increase followed by a drop in activity, and a more rapid subsequent increase” (line 225). Why don’t the authors report such averaged activations across conditions for adult participants as well?

Response:

For the adult participants, the corresponding mean activations are reported in Figure 3A for the scalp analysis and in Figure 4A for the source analysis. As highlighted in the text (lines 193-194), those waveforms show very stereotypical deflections of auditory activity (“Auditory-evoked responses displayed stereotypical N1 and P2 components and were higher in amplitude for spectral than intensity cues.” and (lines 230-234) “The change events evoked neural activity strongly focused on the targeted HG. (Fig. 4A). Both the left and right HG exhibited stereotypical auditory evoked responses for all considered conditions.”). For the newborn participants, however, waveforms of change-evoked auditory responses are less well-known; they are thus also qualitatively described within the main text, as mentioned by the reviewer. Upon a different reviewer’s comment, this description is additionally adjusted to read “The

cluster activations averaged across looming and receding sounds appeared rather shallow until a rapid increase at around 400-500 ms after the event (Fig. 5A)." (lines 279-282).

11. It is not necessary to explain the notation of "(mean \pm standard deviation)" more than once in the text

Response:

We thank the reviewer for pointing out this redundancy: "mean \pm standard deviation" is now only defined upon first mention.

12. For processing software described in 'Additional review materials, Reporting Summary', please specify the Freesurfer version used for MRI segmentation. I believe that for completeness, Sequence and imaging parameters should be reported, together with a specification of whole brain area of acquisition.

Response:

We now specify that Freesurfer [87] v 7.1.1 was used for MRI segmentation and that structural images were acquired using a magnetization-prepared rapid gradient-echo sequence with the following parameters: TE = 2.43 ms; TR = 2300 ms; 208 sagittal slices; field-of-view: 256 \times 256 \times 166 mm; voxel size: 0.8 \times 0.8 \times 0.8 mm. This information is now also included in the manuscript (lines 805 and 677-680)

References cited (by reviewer):

- Chládková, K., Paillereau, N., 2020. The What and When of Universal Perception: A Review of Early Speech Sound Acquisition. *Lang Learn* 70, 1136–1182. <https://doi.org/10.1111/lang.12422>
- Dalboni da Rocha, J.L., Schneider, P., Benner, J., Santoro, R., Atanasova, T., Van De Ville, D., Golestani, N., 2020. TASH: Toolbox for the Automated Segmentation of Heschl's gyrus. *Sci Rep* 10, 3887. <https://doi.org/10.1038/s41598-020-60609-y>
- Destrieux, C., Fischl, B., Dale, A., Halgren, E., 2010. Automatic parcellation of human cortical gyri and sulci using standard anatomical nomenclature. *Neuroimage* 53, 1–15. <https://doi.org/10.1016/j.neuroimage.2010.06.010>
- Granier-Deferre, C., Ribeiro, A., Jacquet, A.Y., Bassereau, S., 2011. Near-term fetuses process temporal features of speech. *DevSci* 14, 336–352. <https://doi.org/10.1111/j.1467-7687.2010.00978.x>
- Querleu, D., Renard, X., Versyp, F., Paris-Delrue, L., Crèpin, G., 1988. Fetal hearing. *European Journal of Obstetrics and Gynecology and Reproductive Biology* 28, 191–212. [https://doi.org/10.1016/0028-2243\(88\)90030-5](https://doi.org/10.1016/0028-2243(88)90030-5)
- Zoellner, S., Benner, J., Zeidler, B., Seither-Preisler, A., Christiner, M., Seitz, A., Goebel, R., Heinecke, A., Wengenroth, M., Blatow, M., Schneider, P., 2019. Reduced cortical thickness in Heschl's gyrus as an in vivo marker for human primary auditory cortex. *Hum Brain Mapp* 40, 1139–1154. <https://doi.org/10.1002/hbm.24434>

REVIEWER #2 (REMARKS TO THE AUTHOR):

Review of COMMSPSYCHOL-23-0286-T by Baumgartner and colleagues

The authors report an interesting and valuable study on a difficult participant group (newborn infants), purportedly demonstrating the presence of a bias in processing for auditory looming (over receding) stimuli in adults and newborns in the intensity change domain, but and a bias for looming in adults but not newborns in the spectral domain. Overall I am positive about this work. However, I have some reservations about the claims the authors make and the level of control they have in the experiments. I cannot recommend publication in the article's current form, not least because of apparent mistakes in reporting.

Rationale and interpretation:

1. The authors indicate that their aim is to disentangle inborn vs learned responses to auditory looming. To do this they present changes in "distance" by means of changes of intensity (accessible to foetuses prenatally) and spectral shape (not accessible prenatally). In newborns the authors they find a looming bias only when using intensity and therefore claim the looming bias has an innate component ("the bias comprises innate components while flexibly incorporating the spatial cues acquired through lifelong exposure"). Because intensity info is accessible prenatally I am not convinced that we can rule out a role for prenatal experience in addition to or instead of phylogenetic factors. This is perhaps less likely given the comparative literature that the authors cite, but I think that this should at least be considered as a possibility. The authors also seem rather fixated on the view of looming biases being related to threat. Perhaps this is because of their phylogenetic stance. However, there is the possibility that differences between looming and receding can be learned via non-threatening but still salient multisensory information both pre and postnatally.

Response:

We thank the reviewer for pointing out potential alternatives for the origins of looming bias. We adjusted the introduction accordingly, clarifying that the threat relationship is hypothetical (lines 39-43):

"The effect of approaching sounds being more salient than receding ones constitutes the "auditory looming bias"; a perceptual bias, presumably present to warn the sensorimotor system to take protective action. Studies corroborate this hypothesized protective nature across vertebrates."

and removed the relevant keywords ("threat detection, innate fear") from the list of keywords below the abstract. We additionally acknowledged the possibility of prenatal learning mechanisms, in the following (lines 427-431):

"Despite corroborative evidence, our findings should be conditional to cautious interpretation. The presence of intensity bias, the most salient cue type, in the newborn brain, is a finding well in line with it having innate components potentially stemming from evolution. It is a possibility, though, that to some degree that bias results from learning during intrauterine sensory development [52]"

Design decisions:

2. Somewhat more problematic for interpretation of the newborn findings, I am not convinced that the authors' effects are due to changes in intensity. In the looming and receding conditions, changes in intensity from soft to loud (looming) vs loud to soft (receding) are perfectly confounded with the intensity of the last stimulus presented (loud for looming and soft for receding). Unless I have missed a clever feature of design or analysis which controls for this, I am concerned that the newborn findings could be due just to the intensity of the last stimulus presented and its continued presence. Perhaps this might explain what the authors do not seem to pick up very defined ERPs from the newborn group (making it difficult for them to undertake the comparisons in peaks employed with adults). Related to this question, it is worth asking whether newborns can notice a 5dB change. Do the authors have evidence to corroborate this design choice?

Response:

We thank the reviewer for their comment. To rule out that biases were induced by absolute sound intensity rather than changes in intensity, we performed the onset analysis (appendix A2). There, we compared the neural signatures of stimuli corresponding to the two simulated positions (near - far) within each stimulus type (spectral and intensity). Should the absolute sound intensity itself be a reason for the signatures we see in the newborns, we would have expected the corresponding time series for the near position (louder/higher intensity) to be, if not higher, then different than the corresponding ones for the far position (soft/lower intensity). As seen in Fig.A2 B and C, no such difference is observed for either stimulus type. In the manuscript this is reported as (lines 292-295):

“we found no significant differences between near and far sounds for either cue type (appendix A), indicating that the observed bias induced by intensity cues is specific to the change event.”

In the revised manuscript (lines 497-511), we also elaborated upon further interpretational uncertainties accompanied by intensity changes, which mainly concerns the newborns, that do not show neural biases with the spectral looming stimuli (that keep the intensity constant).

Regarding the question of whether newborns can notice a 5 dB change, it is known from deviant detection neural responses in oddball paradigms, that the newborn auditory system is capable of detecting quite similarly large changes in intensity (Kostilainen et al., 2018; see also review from Näätänen et al., 2019)

- Kostilainen, K., Wikström, V., Pakarinen, S., Videman, M., Karlsson, L., Keskinen, M., Scheinin, N. M., Karlsson, H., & Huotilainen, M. (2018). Healthy full-term infants' brain responses to emotionally and linguistically relevant sounds using a multi-feature mismatch negativity (MMN) paradigm. *Neuroscience letters*, 670, 110–115. <https://doi.org/10.1016/j.neulet.2018.01.039>
- Näätänen, Risto, Teija Kujala, and Gregory Light, 'The development of MMN,' *The Mismatch Negativity: A Window to the Brain* (Oxford, 2019; online edn, Oxford Academic, 23 May 2019), <https://doi.org/10.1093/oso/9780198705079.003.0003>

Regarding the mentioned difficulty in picking up defined newborn ERPs, we would like to acknowledge the fact, that maturation of the auditory system (i.e., myelin development and plasticity) substantially shapes the generators of acoustic event-related responses. Therefore, the ERP form in the newborn group, i.e. at the beginning of this development process, were expected to differ from adult ERP waves (e.g., Van Den Heuvel et al., 2015; Kushnerenko et al., 2002; Jing et al., 2006; Polver et al., 2023; Bendixen et al., 2015).

- Van Den Heuvel, M. I. *et al.* Differences between human auditory event-related potentials (AERPs) measured at 2 and 4 months after birth. *Int. J. Psychophysiol.* **97**, 75–83 (2015).
- Kushnerenko, E. *et al.* Maturation of the auditory event-related potentials during the first year of life. *NeuroReport* **13**(1), 47–51 (2002).
- Jing, H. & Benasich, A. A. Brain responses to tonal changes in the first two years of life. *Brain Dev.* **28**(4), 247–256 (2006).
- Polver, S., Háden, G.P., Bulf, H. *et al.* Early maturation of sound duration processing in the infant's brain. *Sci Rep* **13**, 10287 (2023).
- Bendixen, A., Háden, G. P., Németh, R., Farkas, D., Török, M., & Winkler, I. (2015). Newborn infants detect cues of concurrent sound segregation. *Developmental neuroscience*, 37(2), 172–181. <https://doi.org/10.1159/000370237>

3. There is also a more subtle matter of interpretation. The intensity change happens within only a 10 ms window. The authors state that the transition phase itself was kept very short (10ms) to assure high temporal precision in the analysis of neural responses evoked by the change event. To what extent is this information really relevant to picking up looming vs receding of real objects in the real world?

Response:

We thank the reviewer for this intriguing question that stimulated the following additions to the discussion section (lines 487-499):

“The nature of auditory looming can be manifold. The implemented stimuli used in the present study comprise transition ramps in the order of 10 ms. The ecological validity of using a 10 ms duration to simulate looming or receding sounds depends on the natural soundscape and the types of events or objects being simulated. In certain real-world scenarios, such rapid changes may not be as common or provide sufficient information for accurate perceptual judgments. In our experimental setting, however, keeping the transition phase short was not only crucial for a good temporal isolation of neural processes but also to maintain consistency and comparability with a highly relevant previous studies [8, 24]. This ensured that the looming bias can be reliably elicited, particularly when utilizing the complementary cue type of spectral shapes.

Abrupt increases in sound intensity may also be judged as salient onset events [54] rather than motion events. While this confounds the interpretation of biases found for the intensity condition, it is less clear for the spectral condition.”

4. Some other questions about rationale, design, and interpretation:

In adults the authors find larger results in the active vs passive condition (i.e. when attention is at play). Then why having the newborns in deep sleep vs e.g. awake or active sleep?

Response:

Adult listeners were tasked to attend to the stimuli and give a response regarding their simulated direction. Fulfilling that task assumes understanding and attentional focus, which an awake newborn listener cannot be tasked with. Newborns are awake for only few hours in a day, and a 30-40 min EEG awake recording in sufficient SNR quality is rather unattainable: They are frequently moving, rendering the EEG data collection and analysis non-trivial due to a large amount of artifacts. Similarly, active sleep is relatively short and the newborn may move, groan, open their eyes, cry out or breathe noisily or irregularly (back-and-forth movement of the eyes under the lids; high levels of brain activity; irregular breathing; and an elevated, but variable, heart rate). In contrast, during quiet sleep, babies lie relatively still. There are no rapid eye movements, and their breathing and heart rates become more regular. Overall, babies in quiet sleep are much less likely to make noise and introduce artifacts. It is therefore a preferable state and common practice for newborn EEG recordings - see also:

- Corsi-Cabrera M, Cubero-Rego L, Ricardo-Garcell J, Harmony T. 2020. Week-by-week changes in sleep EEG in healthy full-term newborns. *Sleep*. 43(4):zsz261.
- Dereymaeker A, Pillay K, Vervisch J, De Vos M, Van Huffel S, Jansen K, Naulaers G. 2017. Review of sleep-EEG in preterm and term neonates. *Early Hum Dev*. 113:87-103.
- Ficca G, Fagioli I, Salzarulo P. 2000. Sleep organization in the first year of life: developmental trends in the quiet sleep-paradoxical sleep cycle. *J Sleep Res*. 9(1):1-4.
- Frank MG. 2020. The Ontogenesis of Mammalian Sleep: Form and Function. *Curr Sleep Med Rep*. 6(4):267-279.

Our motivation is now better defined and included in the

- Results section (lines 268-269): “Apart from feasibility reasons [32-34], the deep sleep state ensured no attentive mechanisms were active.”
- Methods section (lines 629-634): “The sound presentation for newborn listeners was equivalent to the passive condition of the adults and they underwent the experiment while in deep sleep. In contrast to active sleep, this state lasts longer (up to 60-90 min), has no rapid eye movements and the breathing and heart rates of the newborns become more regular. Overall it is a preferable state for EEG recording, as the introduction of artifacts is much less likely [69-71].”

5. In the discussion, the authors suggest that spatial associations are required for spectral looming cues to be learned. And yet, it seems reasonable that intensity associations would be enough given the correlations available. Can they clarify this case?

Response:

We added the following discussion in order to elaborate upon this matter (lines 499-511):

“On the one hand, understanding of the spectral cues is expected to rely on spatial associations [37,55]. Acquiring those associations is therefore thought to facilitate the bias, purely from a spatial point of view and, by stimulus design, without intensity confounds. Nevertheless, associative learning could be possible, meaning that the significance of one cue gets learned purely based on understanding of another. Although isolated in our study design, intensity and spectral cues do not appear as such in nature, therefore obscuring the precise interdependencies. Associative learning may also explain why full motion cues are the most efficient in facilitating the warning mechanism of the looming bias [7]. To further investigate this question, it would be interesting to study the possibility of inducing looming bias with novel spectral cues. Those should have been acquired through directional localisation training and with stimulus intensity being varied to rule out intensity associations [56].”

6. The authors don't explain why they identified a particular sample size or report any power analysis. How was this matter approached?

Response:

The sample size for the adult group was determined based on the following considerations: As of Baumgartner et al. (2017), 15 subjects should be sufficient to detect the looming bias via scalp potentials evoked at latencies of about 160 ms for the active spectral condition. We decided to double the sample size because effect sizes were expected to be smaller under passive listening conditions, because we wanted to allow for finding neural signatures also at shorter latencies (usually of smaller amplitude and therefore harder to discern), and because we are aiming to re-use the data for additional exploratory studies based on functional connectivity analyses, generally requiring higher sample sizes. This motivation is now included in the methods section (lines 528-535).

Regarding the newborn sample size, previous event-related potential studies on neonatal auditory change detection found effects reliably with about 40 subjects (Toth et al., 2023; Van Den Heuvel et al., 2015; Kushnerenko et al., 2002; Jing et al., 2006; Polver et al., 2023; Bendixen et al., 2015). Since, due to the very specific design and paradigm, null results were to be expected in our study, we decided to substantially increase the sample size and recruit at least 100 participants. This motivation is now included in the methods section (lines 554-558) and also in lines 440-443: “The response biases in adults were already diminished in the passive spectral condition, which might be a precursor for an even smaller effect in the corresponding newborn case, rendering its detection particularly challenging.”

- Tóth, B., Velösy, P. K., Kovács, P., Háden, G. P., Polver, S., Sziller, I., & Winkler, I. (2023). Auditory learning of recurrent tone sequences is present in the newborn's brain. *NeuroImage*, 281, 120384.
- Van Den Heuvel, M. I. et al. Differences between human auditory event-related potentials (AERPs) measured at 2 and 4 months after birth. *Int. J. Psychophysiol.* **97**, 75–83 (2015).
- Kushnerenko, E. et al. Maturation of the auditory event-related potentials during the first year of life. *NeuroReport* **13**(1), 47–51 (2002).
- Jing, H. & Benasich, A. A. Brain responses to tonal changes in the first two years of life. *Brain Dev.* **28**(4), 247–256 (2006).
- Polver, S., Háden, G.P., Bulf, H. et al. Early maturation of sound duration processing in the infant's brain. *Sci Rep* **13**, 10287 (2023).
- Bendixen, A., Háden, G. P., Németh, R., Farkas, D., Török, M., & Winkler, I. (2015). Newborn infants detect cues of concurrent sound segregation. *Developmental neuroscience*, 37(2), 172–181. <https://doi.org/10.1159/000370237>

7. There is little in the discussion which qualifies the claim that spectral information is not used by newborns. Given that individual subject HTRFs are not measurable from newborns, this is a limitation which could have led to the null effect compared to the adults. The authors say something about similar variances between spectral and intensity responses, but do not properly explain this.

Response:

We thank the reviewer for pointing out, that this finding is lacking substantiation. We tried to better elaborate on the caveat of semi-individualization of HTRFs in the discussion (lines 454-461):

“Nevertheless, there inevitably were inter-individual differences in the fit of the approximated HTRFs to the true ones. If newborns were sensitive to the spectral cues provided by their true HTRFs and were hindered solely due to insufficient HRTF individualization, the variance in the goodness of fit of the

HRTFs should be reflected in the variance of the measured neural responses to the spectral condition. The variance we observed in the spectral condition is, however, comparable to the one emerging in the intensity one. Altogether, these considerations provide little evidence for acoustic inconsistencies being the underlying cause. “

and further addressed the use of spectral information by the newborns throughout the manuscript, among others specifically adding the following:

Introduction, lines 87-94:

“HG is already developed around the 24th week of gestation and fetal hearing is functional before birth. Sounds, passing through the mother's abdomen and amniotic fluid during development, undergo spectral modification and attenuation. Intensity ramps are already prenatally accessible [22] and evidence suggests that spectral information is also processed [23]. Yet newborns are additionally subject to abrupt changes in the environment postpartum. This substantially affects the characteristics of spectral cues, thereby necessitating a new acquisition or adaptation process.”

Discussion, lines 368-397:

“In spite of the modifications of the sound characteristics taking place in utero during development, frequencies in the range of 100 - 1000 Hz reach the fetus largely unchanged [23,40], enabling the processing of spectral acoustic information at least for this low-frequency range. Yet the environment in utero, comprising liquid and essentially a low-pass filter for sounds, differs substantially from the one a newborn is postnatally exposed to. Especially given that our modifications affected frequencies beyond 1000 Hz, they have likely undergone essential distortion. Offering limited prenatal experience in the new environment, the absence of a spectral bias postnatally could align with the necessity of spatial associations for the spectral cues to be understood. Along these lines, early behavioral studies suggest that infants gradually acquire them during the first 18 months of their life [41,42].”

“ The experimental set-up for the newborns inevitably differs methodologically from the one for the adults, potentially obscuring the result.” (lines 433-434)

“The response biases in adults were already diminished in the passive spectral condition, which might be a precursor for an even smaller effect in the corresponding newborn case, therefore rendering its detection particularly challenging.” (lines 440-443)

8. It's unclear to me why the authors conclude that they newborn source localisation was sufficiently accurate given they replicated the same effects across both PAC and STS (albeit with smaller effects in STS). Was there a significant different in amplitudes between STS and PAC? This seems particularly important given that the authors argue that the looming bias is present in PAC in both adults and infants.

Response:

We thank the reviewer for pointing out this important detail. Following this suggestion and performing cluster-based permutation statistics within hemisphere for the direct comparison between PAC/(HG) and STG, we found no statistically significant difference between said time series. We therefore removed this claim from our manuscript. We instead identified activations of surrounding regions and also elaborated on the potential activation leakage typically observed for EEG source localization. This relevant information is now included in the results section in lines 232- 238:

“The change events evoked neural activity strongly focused on the targeted HG. (Fig. 4A). Both the left and right HG exhibited stereotypical auditory evoked responses for all considered conditions. In addition, we found high activity at more posterior regions (planum temporale), while activations seem to have leaked into the posterior regions of the insular cortex.”

and the discussion in lines 471-486:

“EEG source localisation relies on assumptions on the spread of activity, as the layers of bone and tissue between the cortical surface and the recording electrodes are inaccessible. As such, the process suffers from imprecisions in the allocation of activity to its cortical generators. Due to its large inter-

subject variability, auditory cortex localization is particularly difficult [18] in that respect. We made use of individual anatomical data and results from previous investigations [31] to infer activity from HG, attempting to limit such imprecisions to the most feasible degree. The analyzed inferred activity resembles the sought auditory cortex one, yet there could also be spill-over from secondary auditory regions. Future investigations with more fine-grained parcellations (e.g. TASH [53]) may give better insights on the dissociation of the two. Studies combining EEG with spatially more precise methods, such as MEG, could, moreover, help better study the cortical generators involved in the bias. This study placed the target on the HG, aiming to investigate auditory cortical signatures of the looming bias; yet further whole-brain connectivity studies might aid towards uncovering the larger network at play, including the multiple ROIs previously shown to be implicated in the biased perception of auditory looms.”

Results reporting:

9. In Figure 3 and the related text, there appear to be inconsistencies and errors. The figure (3C) indicates a difference in latency between spectral and intensity, whereas the text suggest a difference between active and passive (lines 159-160). Why does this difference in latency NOT show up in Fig 3A? There seems to be a problem here. It is not entirely clear what Figure 3D is showing. The y axis suggests this is peaks, but then this seems to be difference waves. This is rather unintuitive and confusing.

Response:

We thank the reviewer for pointing out our confusing descriptions, and attempt to clarify:

Figure 3C demonstrates difference waveforms, i.e., looming-receding, per considered condition. The cited text (lines 159-160 of the original submission) refers to a description of the topographies, i.e., Figure 3B. This representation was done in order to acquire an image of the scalp activations, and see whether our choice of the vertex electrode is a justified one. Seeing that it lies in the middle of the clusters, we proceeded by performing the subsequent ERP analysis on it. The difference in latency on the topographies (Fig. 3B) is mentioned in the cited text (“*The time point of maximum bias manifestation within the clusters differed with cue type and attentive state; within the active state, the maximum bias appeared 44 ms later for intensity cues than spectral cues and 34 ms later in the corresponding passive conditions.*”, lines 207-210). The latency difference emerging in Fig. 3C along with the corresponding statistics is described in appendix B. This is now better specified in lines 219-222 (“*Attention showed little effect on N1 peaks but significantly magnified P2 biases, especially for intensity cues. Details of this statistical evaluation along with the corresponding analysis for the latency are provided in appendix B.*”)

Figure 3A shows mean activations, i.e., (looming+receding)/2, per considered condition. This specification is now included in the caption of Figure 3. Aim of this depiction was to demonstrate, that the auditory stimuli overall evoke the expected response. Yet, when considered with regards to the bias evaluation (so looming-receding) the latency differences appear. As described in the caption, Figure 3D is showing the values of the peak amplitudes and the statistics done thereon. The peaks were extracted separately for every trial, condition and subject (i.e. regular ERP waveform) and then the difference in amplitude or latency was calculated to obtain the peak bias metrics plotted in Fig 3D. The ordinate label is now changed to “Peak bias amplitude”, and the caption changed to “*C) Difference waveforms (looming – receding) at the vertex electrode and D) the extracted peak amplitude values of the N1 and P2 components. Statistical significance emerges for the factors of cue type and attention.*”

10. In the adults sample, why are ERPs calculated only on 1 electrode (Cz) as opposed to a cluster (conversely, in newborns they use a fairly large cluster)? This seems a strange decision. Also when have the authors chose to compare peak amplitudes rather than, for instance, mean individual amplitudes given problems with peak amplitudes as a measure?

Response:

Due to the constraints in application (time, ease of application), the EEG setup used for the newborn listeners was different than the one for the adults. The newborn-EEG design is using the Cz electrode as a reference (as denoted in line 691 of the methods section). In infant ERP analysis it is a common practice to use a cluster of EEG channels due to the low SNR (Monroy et al., 2021). Therefore to measure auditory N1 response in newborns we pooled the activity of neighboring EEG channels. This is now better defined in the methods, in lines 874-879:

“In contrast to the EEG recordings in the adult group, Cz was used as a reference during newborn recordings. Following the common practice of infant ERP analysis, a cluster of channels was considered to estimate the effects [92,93]. We calculated the scalp-ERPs based on the emerging frontocentral cluster of electrodes (Fig. 5D), comprising electrodes *Fp1, AF7, AF3, AFz, AF4, F5, F3, F1, Fz, F2, F4, F6, FC3, FC1, FC2, FC4, C1, C2* and *C4*.”

- Monroy, C., Domínguez-Martínez, E., Taylor, B., Marin, O. P., Parise, E., & Reid, V. M. (2021). Understanding the causes and consequences of variability in infant ERP editing practices. *Developmental psychobiology*, 63(8), e22217.

The choice of Cz for the adults was made following the calculated topographies (Fig. 2B) and consistency with Baumgartner et al. (2017), now better motivated in lines 190-192 of the results section.

Regarding the choice of extracting peak amplitude measures rather than mean amplitudes over a certain time window, we would like to point out that we aimed for a procedure that extracts both the amplitude and latency of ERP components. To our knowledge, there are many suggestions on how to extract those measures yet no clear consensus on the best way to do so. An approach that very often is taken but just does not seem reasonable to us is to simply average the activity within a certain time window. From a signal processing point of view, this corresponds to low-pass filtering the waveform with a rectangular temporal window function that causes strong side lobes in the frequency domain. If the purpose of such windowing is to target a specific frequency region, then it would make more sense to use a window function that is more constrained in the frequency domain, as for instance a Gaussian window. “*Since we already low-pass filtered the data at 20 Hz we deliberately opted against additional low-pass filtering through some form of temporal averaging, and simply took the amplitude and latency of peaks identified within certain time windows that are consistent with reported N1 and P2 latencies reported in the literature.*” This last part is now included in the methods section in lines 826-830.

11. Lastly, I feel the authors could do more to clarify the nature of the drift rate latent variable which they extracted from the adult behavioral data.

Response:

We thank the reviewer for pointing out our unclear description. More information has been added:

“We selected this model-based approach because of its advantage in accounting for the speed-accuracy trade-off on a trial-by-trial level [28] as well as the different uncertainty levels across subjects [27]. In this modeling framework, an evidence accumulation process is started for every choice option and trial; the accumulator hitting the response threshold first decides the choice as well as the response time. To study the presence of looming bias, our latent variable of interest was the drift rate, which quantifies the velocity of evidence accumulation towards a response in a forced choice task [29]. With drift rates fitted for every stimulus condition, simulated data demonstrated high agreement with the actual hit rates and response times across all conditions ($p > 0.33$; Fig. 2A).” (lines 167-177)

Detailed comments:

12. The authors, in their EEG analysis claim to be investigating looming biases in “sensor space”, but then proceed to an analysis of just the vertex electrode. Sensor space seems like an incorrect label for this kind of analysis. Just say “In EEG gathered from the scalp”?

Response:

The characterization “sensor space” was used in order to dissociate between the two; sensor- and source space. We would like to note that, apart from the ERP analysis on the vertex electrode, the topographies (Fig. 2B and 5D) and the newborn ERPs are based on signals across all considered electrodes or a cluster thereof – hence, we chose the characterization “sensor space”. We do see the point of the reviewer, though, and changed our terminology from ‘sensor’ to ‘scalp’ consistently throughout the manuscript.

13. Change “no significance” to “no significant difference in xx” throughout.

Response:

We thank the reviewer for their remark; agreed and implemented accordingly.

14. The authors introduced static sounds to ensure that ppts could not predict the stimulus category from the start, however they might have done purely based on the intensity.

Response:

Looming and receding sounds have distinct onset intensities that would allow identification of the sounds purely based on onset and before hearing the change, if no static trials were presented. With static trials such an anticipation is not possible, as one cannot know whether the sound will remain static or transition to a different state. We tried to further clarify this difference in the manuscript:

In the results section (lines 137-140):

“Static sounds were presented in 50% of all cases and served two purposes. First, they ensured listeners were not able to predict the stimulus category already from stimulus onset (appendix A), as static sounds were constructed with the same onsets as the moving stimuli (but no transition).”

In the methods section (stimuli and design, lines 579-586):

“For moving (looming/receding) stimuli, after 600 ± 50 ms, the initial tone complex was crossfaded into the final tone complex using a linear ramp with a duration of 10 ms. Static stimuli, conversely, remained constant throughout.

Moving (looming/receding) and static trials were randomized throughout the experiment and balanced over blocks, with 50% static and 50% moving sounds. Within moving sounds, 50% were looming and 50% receding. Within static sounds, 50% corresponded to the looming stimulus onset and 50% to the receding stimulus onset.”

15. Artifact rejection: excluding channels with artefacts with a criterion of -200:800 microvolt seems excessively lenient for adult data I think?

Response:

It is indeed lenient for adult data and was just kept for the case of some excessive artifact (e.g., a broken channel) was present, that was not noticed in advance. All EEG data were visually inspected at two stages, namely before channel interpolation and after ICA component removal, but it could still happen that large outliers were overlooked. Our choice is now better motivated in the methods of the manuscript (lines 781-785):

“ A hard threshold of -200 to 800 μ V was additionally applied, to detect trials that still had large outlier values, potentially denoting issues that went undetected by visual inspection (e.g., excessive movement artifacts, intermittently broken channels). ”

16. Filtering: .5 is a fairly high high pass, especially for adults (and in fact their infants' high pass is much lower) in newborns

Response:

The cutoff frequency of 0.5 Hz we applied for high-pass filtering is within, but indeed rather at the high end of what is usually recommended in the literature. The earlier the time window of interest, the higher this cutoff can be set without affecting the results (Kappenman & Luck, 2010). Since we targeted relatively early ERP components up to P2, the chosen cut-off should have helped to eliminate slow artifacts while leaving the targeted components largely unchanged. We additionally based our selection on relevant literature (Toth et al., 2020, Szalardy et al. 2021) as well as consistency with the closely related studies of Baumgartner et al. (2017) and Ignatiadis et al. (2021). This motivation is now also included in the methods section, in line 781:

“[...] a threshold chosen to additionally comply with relevant previous studies [8,24].”

Regarding the newborn filtering, we followed specific technical considerations for neonatal EEG to make sure we include the slow oscillations that are typical for neonate brains (Louis et al., 2016, Cherian et al., 2009). This specification is now included in the methods section, lines 856-858.

“Compared to the adults, we chose the highpass cutoff frequency much lower for the newborns to ensure inclusion of the slow oscillations that are typical for neonate brains [92,93].”

- Tóth, B., Honbolygó, F., Szalárdy, O., Orosz, G., Farkas, D., & Winkler, I. (2020). The effects of speech processing units on auditory stream segregation and selective attention in a multi-talker (cocktail party) situation. *Cortex; a journal devoted to the study of the nervous system and behavior*, 130, 387–400. <https://doi.org/10.1016/j.cortex.2020.06.007>
- Szalárdy, O., Tóth, B., Farkas, D., Hajdu, B., Orosz, G., & Winkler, I. (2021). Who said what? The effects of speech tempo on target detection and information extraction in a multi-talker situation: An ERP and functional connectivity study. *Psychophysiology*, 58(3), e13747. <https://doi.org/10.1111/psyp.13747>
- Kappenman, E. S., & Luck, S. J. (2010). The effects of electrode impedance on data quality and statistical significance in ERP recordings. *Psychophysiology*, 47(5), 888–904. doi:10.1111/j. 1469-8986.2010.01009.x

17. "Trials numbers were equalised across conditions within each subject by removing trials equally distributed across the recordings in order to match the minimum amount within the subject". This seems an odd practice given the robustness of parametric analysis to small violations of assumptions. How did the authors decide which trials to remove?

Response:

After all artifact cleaning, the trial equalization was done based on a selection of trials, that ensures both equal amount but also equal sampling across the recorded session (c.f., Baumgartner et al., 2017). More specifically, we selected every y/x -th trial in order to remove x trials from a set of y trials.

This process is now better specified in the methods section (lines 794-798):

“Trials were equalized within each subject to match the minimum amount within the subject after trial rejection, aiming at an equal distribution across the recordings. More specifically, we selected every $(y/x)^{\text{th}}$ trial in order to remove x trials from a set of y trials, rendering the same amount of trials across conditions within a subject”

18. line 443: newborns were presented with 50 trials per condition vs line 653 average of 82 valid trials per condition.

Response:

We thank the reviewer for finding this mistake in our reporting. Newborns were presented with 50 trials per presentation side for each condition, hence 100 trials per condition. Out of those, on average, 82 were valid, following the cleaning process. This was now corrected to read (lines 634-638):

“ In total, the experiment lasted approximately 30 minutes and consisted of 400 trials, with 100 trials per condition (as only the passive condition was considered, in contrast to the adult experiment). Within a condition, 50 trials were each presented from the left and the right side of the listeners.”

REVIEWER #3 (REMARKS TO THE AUTHOR):

Main points

This is an extensive, well-conducted study that presents an impressive amount of work. The large sample sizes, refined source analysis procedure, and advanced statistical methods are particularly noteworthy.

1. In theory, the idea to contrast spectral and intensity cues to distinguish between the learned and innate components of the looming bias, respectively, is indeed compelling. However, even the adult EEG data only showed a relatively small bias in response to spectral cues in the passive condition. My main concern is that the absence of this effect in the newborns cannot simply be attributed to their inability to use these cues, as the newborn experiment inevitably represents a downgrade in almost every methodological regard. For instance, no MRIs and electrode positions could be obtained, the stimuli were template-based rather than individualised, and the newborns were sleeping rather than just inattentive. Hence, it is not entirely clear if there was indeed a difference between groups in that regard or if the data just weren't precise enough to uncover an effect in the newborns. Although some limiting factors are discussed, the authors are trying a bit too hard to sell their results at face value and I would welcome a somewhat more cautious interpretation of the results.

Response:

We thank their reviewer for their constructive comment and agree that the mentioned points should be described in a more salient manner. We would like to point out, however, that the sample size was substantially increased for the newborn group and that the lack of individual brain anatomies and electrode positions only compromises the results obtained for the source-localized analysis and not for the one at the scalp. Hence, we mainly focus our discussion on the state of consciousness and incomplete individualization of spectral cues, as well as the potential experience of newborns with spectral information. Acknowledging the need for more cautious interpretation of our results, we elaborate on that in our (new) section of limitations and future directions. Among others, the manuscript has now been changed, to include:

“Although not found in our setting, the presence of spectral associations in the newborns' brains cannot be entirely refuted. The experimental set-up for the newborns inevitably differs methodologically from the one for the adults, potentially obscuring the result.” (lines 431-434)

“A possible reason for not finding an existing effect concerns the newborns' state of consciousness. We compared passively listening adults to sleeping newborns, a standard procedure in cross-age auditory research [32-34]. Although it has been demonstrated that awake and sleeping newborns show identical neural responses to sounds and changes in sound properties [49], the generalization of this to cue-specific looming sensation may come with some uncertainty. The response biases in adults were already diminished in the passive spectral condition, which might be a precursor for an even smaller effect in the corresponding newborn case, rendering its detection particularly challenging.” (lines 440-443)

“Nevertheless, there inevitably were inter-individual differences in the fit of the approximated HRTFs to the true ones. If newborns were sensitive to the spectral cues provided by their true HRTFs and were hindered solely due to insufficient HRTF individualization, the variance in the goodness of fit of the HRTFs should be reflected in the variance of the measured neural responses to the spectral condition. The variance we observed in the spectral condition is, however, comparable to the one emerging in the intensity one. Altogether, these considerations provide little evidence for acoustic inconsistencies being the underlying cause.” (lines 454-461)

“Despite all taken measures and the more than twofold sample size of the newborn group, the possibility of them not sufficiently counterbalancing the imposed methodological limitations has to be acknowledged.” (lines 467-469)

Additionally, limitations regarding the stimuli construction as well as the precision of the cortical source localization have been acknowledged and discussed (discussion section, lines 470-486 and 487-511).

2. Although quite some effort has been taken to conduct the source analyses (individual MRIs and channel positions in adult sample) and the use of the dSPM algorithm resulted in very fine-grained distributed source reconstructions, the discussion of the cortical generators involved remained very superficial and their anatomical description is mostly inadequate. Please see comments re Lines 183, 186, 241, and 615 below. The authors should update the relevant sections by correctly naming the different areas that showed activity and describe which processing steps they are commonly associated with, particularly with respect to PAC and planum temporale.

Response:

We thank the reviewer for their detailed comments on these shortcomings; the lines in question were addressed (see below). We now better clarify that we are focusing on HG as the anatomical region of interest, housing the primary auditory cortex, and better describe the activated regions.

Specific comments:

3. Abstract: This needs a little more content. What exactly did the source localisations reveal in your EEG data?

Response:

Agreed. We extended the abstract to include:

“Through cortical source localisation we demonstrate the emergence of the bias in both age groups at the level of Heschl's gyrus. Adults demonstrate the bias in both attentive and inattentive states; yet differences in amplitude and latency appear based on attention and cue type. Contrary to the adults, in newborns the bias was elicited only through manipulations of intensity and not spectral cues.”

[Please note that the above change is not highlighted in the annotated version of the manuscript.]

4. Line 68: Too vague, please describe these findings briefly.

Response:

The manuscript has now been updated to read (lines 70-77):

“The specific involvement of the auditory cortex is, however, surprisingly obscure in those: Its appearance as an important contributing region is inconsistent across studies and raises the need for more investigations targeted towards it, under consideration of different human brain states. Apart from that, localisation of the auditory cortex in neuroimaging studies is non-trivial: although the medial part of the anatomical region of Heschl's gyrus (HG) is generally considered to host the primary auditory cortex, it remains a functional definition suffering large inter-subject variability [18].”

5. Line 109: This is probably too short to even notice the jitter.

Response:

In event-related potential (ERP) studies, it is generally recommended to temporally jitter the event of interest by at least 50 ms because then trial averaging minimizes the effects of anticipatory neural processes, as well as the undesired influence of alpha waves due to vigilance (Luck, 2005). It is therefore not required, that study participants notice the jitter consciously.

– Luck, S. J. (2005). An introduction to the event-related potential technique. MIT Press.

6. Fig. 1: The lower panel of B is hard to grasp and requires more explanation. For the three signals with a non-flat spectral envelope, it looks as if you just altered the overall power, i.e., DC offset.

Response:

We thank the reviewer for motivating a better explanation of our stimuli. We constructed stimuli that either had the intensity or the spectral attributes unchanged. The blue signals throughout Figure 1 correspond to the HRTF-filtered intensity stimuli, that maintained their spectral characteristics and only changed in intensity (hence, DC offset). In contrast to the intensity stimuli, the spectral cues (red curves) maintained

their intensity level over time (flat lines in Fig. 1B, top) but changed their spectral characteristics (from flat to HRTF-filtered stimulus, Fig.1B bottom). The manuscript was changed to incorporate this essential explanation, to (lines 120-130):

“Moving sounds differed from static sounds by having a brief cue transition phase about halfway through the stimulus (Fig. 1B, top, grey area represents the transition phase in time). The movement percept for our stimuli was created by changing either the intensity (Fig. 1B, top, blue) or the spectral shape (Fig. 1B, bottom, red curves) of a broadband harmonic tone complex. In the first case (intensity stimuli, blue curves) the intensity changed with time (Fig. 1B, top, blue) while the spectral content remained the same (Fig. 1B, bottom, blue), essentially representing a mere offset. Spectral stimuli maintained their overall intensity in time (Fig. 1B, top, red), but transitioned in spectral content between a flat spectrum and the measured HRTF (Fig.1B, bottom, red).”

7.Line 132: Please back this up with some stats.

Response:

We performed the requested statistical tests and now reported (lines 156-160 of the results section):

“When comparing to static sound detection with the Wilcoxon Signed-Rank test, the discrimination of movement direction in motion trials was substantially harder (hit rates: [0.514, 0.648, 0.757] - $V = 406$, $N = 27$, $p < .001$) and slower (response times: 1.017 ± 0.170 s - $V = 36$, $N = 27$, $p < .001$)” .

8.Line 178: I don't see what this additional analysis adds to the manuscript. You are merely measuring an unspecific sound onset response here (e.g., Krumbholz et al. 2003, Cerebral Cortex), so it is not surprising that there were no differences between conditions.

Response:

The aim of this analysis was to see whether there were significant differences based mainly on the intensity onsets. It has been discussed, that neural responses can be locked to intensity values, thereby generating higher responses for higher intensity (Teghtsoonian et al., 2005). This poses a potential confound in intensity studies, and could mean that the difference we define as bias is actually just the reflection of the differences in intensity (higher start intensity for receding vs. lower start intensity for looming). Through our onset analysis (Appendix A2) we provide counter-evidence for this confound, through comparing the neural signatures of stimuli corresponding to the two simulated positions (near - far) within each stimulus type (spectral and intensity). If the absolute sound intensity itself was a reason for the signatures emerging in the newborns, we would have expected the corresponding time series for the near position (louder/higher intensity) to be, if not higher, then different than the corresponding ones for the far position (soft/lower intensity). As seen in Fig.A2 B and C, no such difference is observed for either stimulus type. The reasoning for this analysis is now mentioned:

“We found no significant differences within cue type (appendix A), suggesting that the above effects are specific to the change event rather than other sustained stimulus properties like absolute sound intensity.” (lines 225-228)

“ As for the adults, we found no significant differences between near and far sounds for either cue type (appendix A), indicating that the observed bias induced by intensity cues is specific to the change event.” (lines 292-295)

- Teghtsoonian, R., Teghtsoonian, M., & Canévet, G. (2005). Sweep-induced acceleration in loudness change and the “bias for rising intensities”. *Perception & Psychophysics*, 67, 699-712.

9. Line 183: You extracted source waveforms from entire Heschl's gyrus, so this title doesn't apply. Please see comment Line 615.

Response:

This is true; we modified the characterization across the manuscript, and implemented this remark by changing the referenced section title to read (line 229): “*Source activity: Early preattentive bias in Heschl's gyrus.*”

10.Line 186: This is a gross misdescription of the results. Activity was evident around PAC and in planum temporale, rather than the neighbouring STG, and leaked into the insula. The latter effect is typical for EEG source reconstructions and demonstrates their imprecision.

Response:

We thank the reviewer for pointing out the inconsistencies in the description, and attempt to clarify: We based our description on the used atlas, namely the Desikan-Killiany one. In contrast to other atlases, such as the one from Destrieux, the selected atlas does not provide a dedicated parcellation for the planum temporale. It is, instead, part of the defined superior temporal gyrus.

Desikan-Killiany parcellation:

Purple: transverse temporal cortex

Cyan: superior temporal gyrus

Yellow: insular cortex

Destrieux parcellation:

Blue: transverse temporal gyrus

(temp sup-G T transv)

Red: transverse temporal sulcus

(S temporal transverse)

Green: Planum temporale *(G temp sup Plan tempo)*

Black: Posterior ramus of the lateral sulcus

(Lat Fis-post)

Yellow: Inferior segment of the circular sulcus

of the insula (S circular insula inf)

Magenta: Lateral aspect of the superior

temporal gyrus (G temp sup Lateral)

Beige: Short insular gyru *(G insular short)*

Cyan: Superior segment of the circular sulcus

of the insula (S circular insula sup)

Orange: Long insular gyrus and central sulcus

of the insula (G Ins lg and S cent ins)

Light green: Planum polare of the superior

temporal gyrus (G temp sup Plan polar)

We would like to point out that our choice for the Desikan-Killiany atlas, instead of one providing a finer parcellation, is motivated through the expected imprecision in EEG-based source reconstruction as well as, mainly, for comparability reasons with existing relevant literature (Bidelman et al., 2020; Ignatiadis et al., 2021; Ignatiadis et al., 2022). We nevertheless agree that our current description is misleading and have changed the manuscript accordingly (lines 232-239), to read:

“The change events evoked neural activity strongly focused on the targeted HG. (Fig.4A). Both the left and right HG exhibited stereotypical auditory evoked responses for all considered conditions. In addition, we found high activity at more posterior regions (planum temporale), while activations seem to have leaked into the posterior regions of the insular cortex. Further investigation of these ROIs outside HG was out of scope of the current study.”

We additionally further specified the anatomical characterization of the selected region, in lines 406-411 of the discussion section:

“HG lies on the superior surface of the temporal lobe and functionally houses the primary auditory cortex (Brodmann areas 41 and 42). As defined by the Desikan-Killiany atlas, where it is denoted as transverse temporal gyrus, it comprises the area between the rostral extent of the transverse temporal sulcus and the caudal portion of the insular cortex. The lateral fissure and the superior temporal gyrus are the medial and lateral boundaries, respectively [49].”

and commented on the mentioned imprecision that comes with cortical source localization through EEG in lines 471-486:

“EEG source localisation relies on assumptions on the spread of activity, as the layers of bone and tissue between the cortical surface and the recording electrodes are inaccessible. As such, the process suffers from imprecisions in the allocation of activity to its cortical generators. Due to its large inter-subject variability, auditory cortex localisation is particularly difficult [18] in that respect. We made use of individual anatomical data and results from previous investigations [31] to infer activity from HG, attempting to limit such imprecisions to the most feasible degree. The analyzed inferred activity resembles the sought auditory cortex one, yet there could also be spill-over from secondary auditory regions. Future investigations with more fine-grained parcellations (e.g. TASH [53]) may give better insights on the dissociation of the two. Studies combining EEG with spatially more precise methods, such as MEG, could, moreover, help better study the cortical generators involved in the bias. This study placed the target on the HG, aiming to investigate auditory cortical signatures of the looming bias; yet further whole-brain connectivity studies might aid towards uncovering the larger network at play, including the multiple ROIs previously shown to be implicated in the biased perception of auditory looms.”

11.Line 193: These waveforms look a lot noisier than the scalp ERPs and the pattern in the results is also less clear cut. It seems too much a simplification to just state that the bias elicited by spectral cues has a shorter latency.

Response:

The mentioned characterization refers to a qualitative description of the curves depicted in Figure 4B. For that reason, we modified the text to read (lines 241-245):

“ In both cortices, we observed qualitatively similar waveforms, that were also congruent to the scalp responses (Fig. 3C): Active conditions seemed to exhibit larger looming biases than passive conditions, and biases elicited by spectral cues appeared smaller and earlier to those elicited by intensity ones.”

To investigate these claims we statistically investigated the curves by means of peak bias amplitude and latency. The described latency difference appeared statistically significant for the factor of cue type, for both the N1 and P2 peaks, which is what prompted the quoted by the reviewer statement in our manuscript. The relevant information is provided in Appendix B and Figure B3.

12.Line 226: To me these ERPs look flat until about 500 ms after the acoustic change.

Response:

We acknowledge the point raised by the reviewer and changed the description in the text, to read:

“The cluster activations averaged across looming and receding sounds appeared rather shallow until a rapid increase at around 400-500 ms after the event (Fig. 5A).” (lines 279-282)

13.Line 241: There is quite a bit of activity outside auditory cortex that should be mentioned too.

Response:

We thank the reviewer for pointing out this omission. This activity is now commented upon, in lines 231-305 of the results section:

“The change events also evoked activity in more distributed cortices of the newborns, including the superior and inferior temporal gyrus and occipital area. These observed activations might be attributed to object movement initiating rapid multisensory associative cortical processes, or the role of sleep in newborns' sensorimotor development [35]”

14.Line 485: This raises the possibility that spectral cues were per se more difficult to detect for newborns as a template HRTF was used.

Response:

This is indeed a limitation of our study and is now more acknowledged and elaborately commented; the relevant segments are mentioned in the response to comment #1.

15.Line 615: PAC is not the same as the transverse temporal gyrus, i.e., Heschl's. The medial part is generally considered PAC, while the anterolateral part contains secondary areas.

Response:

We thank the reviewer for their constructive comment. We recognize our mistake and have adjusted the characterization throughout the manuscript in the following manner.

- Addition to the introduction, to better describe and dissociate the regions (lines 73-77): “Apart from that, localization of the auditory cortex in neuroimaging studies is non-trivial: although the medial part of the anatomical region of Heschl's gyrus (HG) is generally considered to host the primary auditory cortex, it remains a functional definition suffering large inter-subject variability [18].”
- Removing the functional characterization of “PAC” from the results section and only referring to the actual anatomical regions in question, namely Heschl's gyrus (HG) throughout the results section.
- Addition to the discussion (lines 406-411): “HG lies on the superior surface of the temporal lobe and functionally houses the primary auditory cortex (Brodmann areas 41 and 42). As defined by the Desikan-Killiany atlas, where it is denoted as transverse temporal gyrus, it comprises the area between the rostral extent of the transverse temporal sulcus and the caudal portion of the insular cortex. The lateral fissure and the superior temporal gyrus are the medial and lateral boundaries, respectively [49].”

16.Line 620: Using the grand mean across subjects and then specifying time windows based on this would have been a more convincing approach.

Response:

We tried the suggested approach initially and also obtained meaningful results for most cases. Two representative examples are shown below, where the asterisks denote peaks selected as N1 (left panel) or P2 (right panel):

However, in a few cases with less pronounced deflections, the variability in peak latency was too high; it resulted in either yielding no peaks or getting trapped in local minima/maxima in return. An example of such a case is shown below (orange curve, transverse temporal L): The left plot shows the result of the grand mean, while the right plot the result from the adjustment.

These adjustments in the selection of the timing did not affect the vast majority of cases. Yet in order to proceed with the statistical analyses, we would have had to either exclude the seemingly problematic cases, or extrapolate the values based on the remaining ones. After visual inspection of the problematic cases, we concluded that a general, small adjustment to accommodate most “outliers” is feasible, in return for less arbitrary or missing values. It is additionally noted, as described in the methods, that we still had some missing values, resulting from an inter-subject variability we could not rationally accommodate by adjusting our selection. Those cases were excluded from the analyses, as had also been described in the methods, lines 838-843:

“In cases where no peaks could be found, such as for poor source localisation or untypical scalp timeseries profiles lacking peaks, the corresponding subjects were not considered in the statistical analyses (concerned 2 subjects each for scalp P2, source N1, and source P2). We opted for that solution as it was deemed a more objective one, compared to arbitrarily assigning a peak value based on literature values or subject means.”

17. Line 641: The high-pass cutoff at 0.05 Hz is much lower than the one for the adults (0.5 Hz). There appear to be strong drifts in the newborn ERPs. Could this be due to the low cutoff?

Response:

We thank the reviewer for pointing out that we omitted the explanation of how we chose the cutoff frequency of the high-pass filter. We tried to follow specific technical considerations for neonatal EEG to make sure we include the slow oscillations (0.2 - 0.5 Hz) that are typical for neonate brains (Cherian et al., 2009). Specifically, they suggest to set the cutoff at 0.005–0.01 Hz to make sure to include those slow

oscillations but at the same time warn about the fact that this very low filter setting may make the EEG more vulnerable to slow artifacts like sweat potentials, as hinted at by the reviewer. We thus decided that setting the cut-off of our high-pass filter at 0.05 Hz may be a good trade-off between sufficient artifact removal and maintaining the desired neural activity. This motivation and reference is now included in the methods section, lines 856-858.

- Cherian, P. J., Swarte, R. M., & Visser, G. H. (2009). Technical standards for recording and interpretation of neonatal electroencephalogram in clinical practice. *Annals of Indian Academy of Neurology*, 12(1), 58. <https://doi.org/10.4103%2F0972-2327.48869>

18.Line 670: Why this time window?

Response:

As there were no specific peaks that would allow us to exactly follow the statistical process we followed in the adult data, we chose this time window as representative of the potential looming activation, based on Fig. 6 B (cluster of intensity bias). As this choice is intrinsically vague, we attempted the test with different time windows, which yielded no difference in the results. We now specified this motivation in the text (lines 888-901):

“To that end, we averaged the corresponding HG time series across the time interval between 250 and 450 ms in the looming as well as receding spectral cue time series . As there were no specific peaks that would allow us to exactly follow the statistical process we followed in the adult data, we chose this time window as representative of the looming bias activation, based on the significant clusters found for the intensity condition (Fig. 6). We investigated the effects across matched models using default settings (r scale fixed effects = 0.5, r scale random effects = 1, r scale covariates = 0.35). As the choice of this time interval is to some degree arbitrary, we performed robustness tests by repeating the same procedure for an earlier time interval (200 - 400 ms), as well as for the latest interval of 600-800 ms, qualitatively showing the biggest deviation between the looming and receding time series. Changing the considered time windows did not change our results. This analysis was implemented in JASP, version 0.17.3 [96]. ”

4th Mar 24

Dear Dr Baumgartner,

Thank you for your patience during the peer-review process. Your manuscript titled "Cortical signatures of auditory looming bias show exposure-based adaptation across the human lifespan" has now been seen by 2 of the original reviewers, and I include their comments at the end of this message. They are pleased with the revised manuscript and have only minor comments. We are interested in the possibility of publishing your study in Communications Psychology, but would like you to alleviate one final concern, which had been raised by the third reviewer.

In the manuscript, you provide an onset analysis as evidence that ERP differences emerge around the change in tone, and are not are not due to any confound present at the onset of the stimulus:

"we found no significant differences between near and far sounds for either cue type (appendix A), indicating that the observed bias induced by intensity cues is specific to the change event."

Since this is a null effect, we request that you provide positive evidence for the lack of a confound by reporting Bayes Factors or Equivalence Tests.

We therefore invite you to revise and resubmit your manuscript. Please highlight all changes in the manuscript text file.

I am attaching an Editorial Requests Table that details critical reporting requirements for the revised manuscript. Please attend to each item and ensure your manuscript is fully compliant. We are requesting that your manuscript aligns with these requirements as this facilitates the evaluation of your manuscript, reducing delays in re-review and potential future acceptance. If your revised manuscript is not aligned with these requests on major issues, such as those concerning statistics, it may be returned to you for further revisions without re-review. Additional information can be found in our style and formatting guide Communications Psychology formatting guide.

Please use the following link to submit your

- revised manuscript,
- point-by-point response to the referees' comments,
- cover letter (as a separate document),
- the Editorial Policy Checklist (see below),
- the Reporting Summary (see below), and
- the completed Editorial Request Table (attached):

[link redacted]

Best regards,

Kate Storrs

Katherine Storrs, PhD
Editorial Board Member
Communications Psychology
orcid.org/0000-0001-9573-8654

REVIEWER EXPERTISE:

Reviewer #1: neurological auditory development

Reviewer #2: audition, EEG

REVIEWER REPORTS:

Reviewer #1 (Remarks to the Author):

Thank you for addressing my previous comments. I gave no further remarks besides the request to include a reference for the statements that HG is already developed around the 24th week of gestation (e.g., Ghio et al. 2021).

Reference cited:

Ghio, Marta, Cristina Cara, and Marco Tettamanti. "The Prenatal Brain Readiness for Speech Processing: A Review on Foetal Development of Auditory and Primordial Language Networks." *Neuroscience and Biobehavioral Reviews* 128, no. March (2021): 709–719.

Reviewer #3 (Remarks to the Author):

The authors have responded satisfactorily to all points raised and I have no further objections to the publication of this manuscript.

EDITORIAL POLICIES

We ask that you ensure your manuscript complies with our editorial policies and reporting requirements.

To that end, we require revised manuscripts to be accompanied by two completed items: a reporting summary that collects information on study design and procedure, and an editorial policy checklist that verifies compliance with all required editorial policies.

- Nature Research Reporting Summary
- Editorial Policy Checklist

All points on the policy checklist must be addressed. Your revised manuscript can only be sent back to the referees if these checklists are completed and uploaded with the revision.

Notes: If you have submitted a Stage 1 Registered Report, Review, Primer, Comment, or Perspective you do not need to submit these forms. If you have already submitted these forms, you may disregard this request.

* TRANSPARENT PEER REVIEW: Communications Psychology uses a transparent peer review system. This means that we publish the editorial decision letters including Reviewers' comments to the authors and the author rebuttal letters online as a supplementary peer review file. However, on author request, confidential information and data can be removed from the published reviewer reports and rebuttal letters prior to publication. If your manuscript has been previously reviewed at another journal, those Reviewers' comments would not form part of the published peer review file.

Communications Psychology is committed to improving transparency in authorship. As part of our efforts in this direction, we are now requesting that all authors identified as 'corresponding author' create and link their Open Researcher and Contributor Identifier (ORCID) with their account on the Manuscript Tracking System prior to acceptance. ORCID helps the scientific community achieve unambiguous attribution of all scholarly contributions. You can create and link your ORCID from the home page of the Manuscript Tracking System by clicking on 'Modify my Springer Nature account'

and following the instructions in the link below. Please also inform all co-authors that they can add their ORCIDs to their accounts and that they must do so prior to acceptance.
<https://www.springernature.com/gp/researchers/orcid/orcid-for-nature-research>

If you experience problems in linking your ORCID, please contact the Platform Support Helpdesk.

Note: Authors' responses are formatted in green.

REVIEWER EXPERTISE:

Reviewer #1: neurological auditory development

Reviewer #2: audition, EEG

REVIEWER REPORTS:

Reviewer #1 (Remarks to the Author):

Thank you for addressing my previous comments. I gave no further remarks besides the request to include a reference for the statements that HG is already developed around the 24th week of gestation (e.g., Ghio et al. 2021).

Reference cited:

Ghio, Marta, Cristina Cara, and Marco Tettamanti. "The Prenatal Brain Readiness for Speech Processing: A Review on Foetal Development of Auditory and Primordial Language Networks." *Neuroscience and Biobehavioral Reviews* 128, no. March (2021): 709–719.

Thank you again for your valuable feedback in the previous round and the suggested reference that it is now cited in the revised version.

Reviewer #3 (Remarks to the Author):

The authors have responded satisfactorily to all points raised and I have no further objections to the publication of this manuscript.

Once more, we appreciate the valuable feedback received in the previous iteration.

10th Apr 24

Dear Dr Baumgartner,

Your manuscript titled "Cortical signatures of auditory looming bias show exposure-based adaptation across the human lifespan" has now been evaluated by our editorial team. I am delighted to say that we are happy, in principle, to publish a suitably revised version in Communications Psychology under the open access CC BY license (Creative Commons Attribution v4.0 International License).

We therefore invite you to revise your paper one last time to address the remaining concerns of our reviewers and a list of editorial requests. At the same time we ask that you edit your manuscript to comply with our format requirements and to maximise the accessibility and therefore the impact of your work.

EDITORIAL REQUESTS:

SUBMISSION INFORMATION:

OPEN ACCESS:

Communications Psychology is a fully open access journal. Articles are made freely accessible on publication under a CC BY license (Creative Commons Attribution 4.0 International License). This license allows maximum dissemination and re-use of open access materials and is preferred by many research funding bodies.

For further information about article processing charges, open access funding, and advice and support from Nature Research, please visit <https://www.nature.com/commspsychol/article-processing-charges>

At acceptance, you will be provided with instructions for completing this CC BY license on behalf of all authors. This grants us the necessary permissions to publish your paper. Additionally, you will be asked to declare that all required third party permissions have been obtained, and to provide billing information in order to pay the article-processing charge (APC).

* TRANSPARENT PEER REVIEW: Communications Psychology uses a transparent peer review system.

On author request, confidential information and data can be removed from the published reviewer reports and rebuttal letters prior to publication. If you are concerned about the release of confidential data, please let us know specifically what information you would like to have removed. Please note that we cannot incorporate redactions for any other reasons.

* CODE AVAILABILITY: All Communications Psychology manuscripts must include a section titled "Code Availability" at the end of the methods section. We require that the custom analysis code supporting your conclusions is made available in a publicly accessible repository at this stage; please choose a repository that generates a digital object identifier (DOI) for the code; the link to the repository and the DOI must be included in the Code Availability statement. Publication as Supplementary Information will not suffice.

* DATA AVAILABILITY:

[link redacted]

Best regards,

Jennifer Bellingtier

Jennifer Bellingtier, PhD
Senior Editor
Communications Psychology

Katherine Storrs, PhD
Editorial Board Member
Communications Psychology
orcid.org/0000-0001-9573-8654